# Advanced CMOS manufacturing of superconducting qubits on 300 mm wafers

J. Van Damme[1,2], S. Massar[1], R. Acharya[1], Ts. Ivanov[1], D. Perez Lozano[1], Y. Canvel[1], M. Demarets[1,2], D. Vangoidsenhoven[1], Y. Hermans[1], J. G. Lai[1], A. M. Vadiraj[1], M. Mongillo[1], D. Wan[1], J. De Boeck[1,2], A. Potočnik[1✉] & K. De Greve[1,2]

The development of superconducting qubit technology has shown great potential for the construction of practical quantum computers[1,2]. As the complexity of quantum processors continues to grow, the need for stringent fabrication tolerances becomes increasingly critical[3]. Utilizing advanced industrial fabrication processes could facilitate the necessary level of fabrication control to support the continued scaling of quantum processors. However, at present, these industrial processes are not optimized to produce high-coherence devices, nor are they a priori compatible with the approaches commonly used to make superconducting qubits. Here we demonstrate superconducting transmon qubits manufactured in a 300 mm complementary metal–oxide–semiconductor (CMOS) pilot line using industrial fabrication methods, with resulting relaxation and coherence times exceeding 100 μs. We show across-wafer, large-scale statistics of coherence, yield, variability and ageing that confirm the validity of our approach. The presented industry-scale fabrication process, which uses only optical lithography and reactive-ion etching, has a performance and yield in line with conventional laboratory-style techniques utilizing metal lift-off, angled evaporation and electron-beam writing[4]. Moreover, it offers the potential for further upscaling through three-dimensional integration[5] and more process optimization. This result marks the advent of an alternative and new, large-scale, truly CMOS-compatible fabrication method for superconducting quantum computing processors.

In the pursuit of quantum computational advantage and, eventually, fault-tolerant, error-corrected, quantum hardware, a need for more and better physical qubits with high-fidelity control is apparent. Advances in error-correcting codes[6] and quantum gate fidelities could reduce the required number of physical qubits. Additionally, increased stability and uniformity of the qubits would reduce the significant control and tuning overhead[7]. However, for practical applications, the number of physical qubits on a quantum computer will most probably still scale beyond a million[4].

Superconducting circuit implementations of quantum bits have leveraged the scalable nature of solid-state fabrication and have shown tremendous progress in terms of qubit coherence times[4,8] and gate fidelities[9]. State-of-the-art demonstrations include error correction[10–14], processors with hundreds of interconnected qubits and initial claims of quantum supremacy and utility[1,2]. These demonstrations have all been done with architectures utilizing transmon-style qubits[15] with $Al/AlO_x/Al$ Josephson junctions (JJs)[16–18], which are consistently fabricated using angled shadow evaporation and metal lift-off. The advantage of this fabrication technique is the possibility for in situ fabrication of the JJ and minimal etch damage to achieve high-coherence qubits of up to hundreds of microseconds[19–22].

With the scaling requirements of future quantum processors in mind, the industrial-scale fabrication of high-coherence qubits utilizing only all-optical lithography and reactive-ion etching of 300-mm-diameter wafers is an attractive alternative, as it could fully leverage the state of the art in advanced complementary metal–oxide–semiconductor (CMOS) fabrication, in line with recent developments in silicon quantum-dot qubit fabrication[23,24]. This is especially true in view of the enormous sophistication and process control achievable in modern industrial semiconductor tooling sets, as well as the knowledge of advanced three-dimensional integration techniques developed for 300-mm-diameter wafers in the pursuit of Moore's law[25]. Although CMOS foundry-compatible processes[26–28] and qubits fabricated partially in a foundry environment[29] have been shown previously, the fabrication of 300 nm wafer-scale superconducting qubits has not been demonstrated.

In this work, we demonstrate superconducting transmon qubits fabricated on 300 mm silicon wafers in the foundry-standard clean room of the Interuniversity Microelectronics Centre (Imec) using industry-standard methods that leverage earlier learnings[30]. We validated our approach with an extensive across-wafer analysis and benchmarking, for which we characterized 400 qubits and 12,840 JJ test structures, finding excellent qubit yields, qubit coherence times, qubit frequency variability and ageing statistics. Our initial result showcases a leap forward in the potential fabrication volume and yield of high-coherence superconducting qubits, which, together

[1]Imec, Leuven, Belgium. [2]Department of Electrical Engineering (ESAT), KU Leuven, Leuven, Belgium. ✉e-mail: anton.potocnik@imec.be

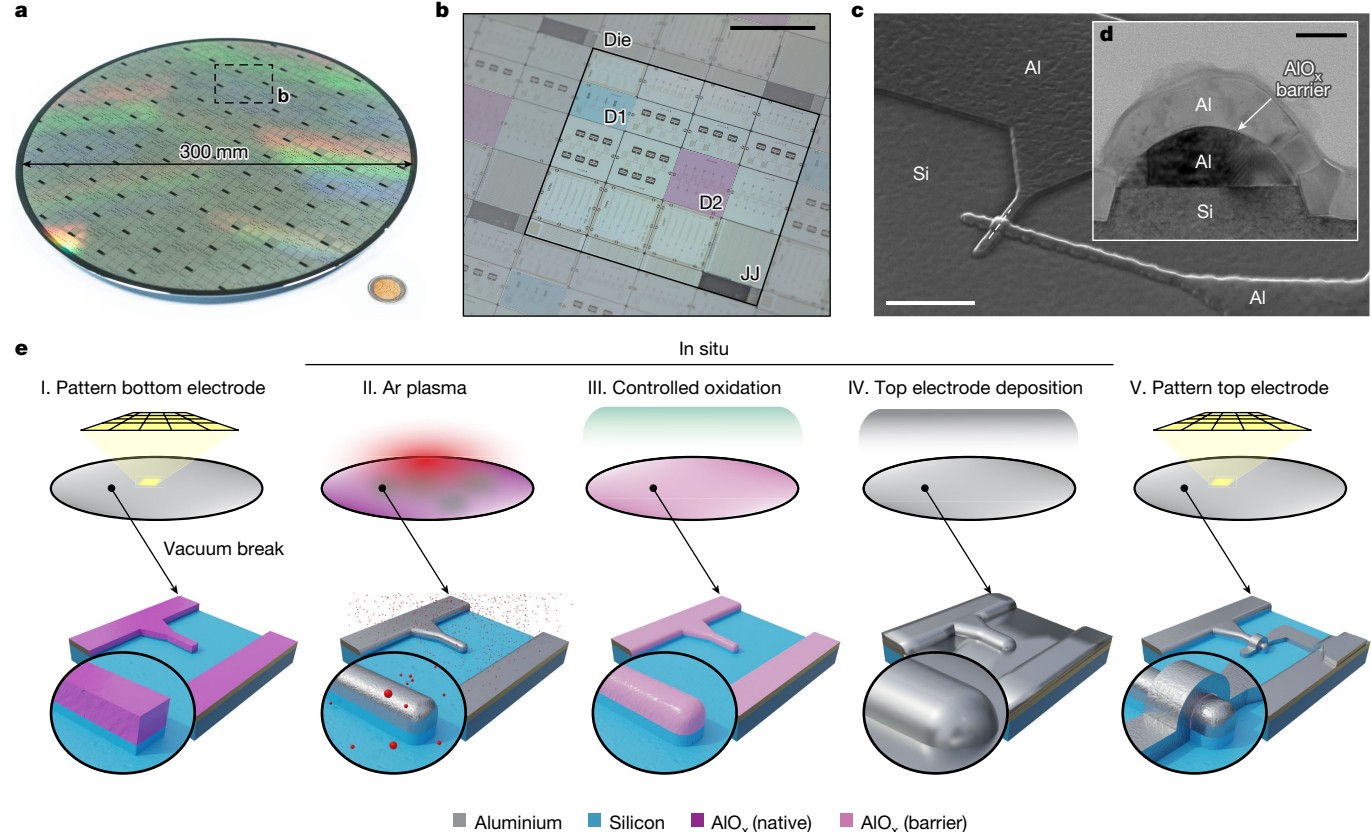

**Fig. 1 | Fabrication of overlap JJ qubit. a**, Photograph of the 300 mm wafer. **b**, Photograph of one die with subdie designs D1 and D2 highlighted. **c**, SEM image of an overlap JJ. **d**, Transmission electron microscopy image of a cross section of the junction (dashed line in **c**). **e**, Schematic representation of the key fabrication steps for an overlap JJ. Scale bars, 10 mm (**b**), 1 μm (**c**), 50 nm (**d**).

Legend: Aluminium | Silicon | $AlO_x$ (native) | $AlO_x$ (barrier)

with three-dimensional integration developments[5,31], could meet the stringent fabrication demands of a future million-qubit processor.

## Design and fabrication

In the industry-standard 300 mm process, an optical mask delineated the pattern of a single die (24 mm × 28 mm), which was replicated 75 times to constitute the complete 300-mm-diameter wafer, as illustrated in Fig. 1a. Each individual die encompassed 20 subdies in our mask, featuring distinct designs of qubits, resonators and JJ test arrays, as shown in Fig. 1b. To assess the quality of the qubits and the variability associated with the fabrication process, parameters such as the normal resistance ($R_n$) of the JJ, qubit transition frequencies ($f_{qb}$), relaxation times ($T_1$) and Hahn-echo coherence times ($T_2^e$) were tracked across the wafer. A subdie design (D1) with five transmon qubits of four different capacitor geometries was selected for the qubit coherence and energy relaxation time analysis. Another subdie design (D2) containing ten qubits with identical capacitors and different JJ areas was selected to monitor the qubit frequency variability. More information on the device designs can be found in Supplementary Information.

The JJ, a crucial element of a superconducting qubit, was fabricated using a 300 mm technology compatible overlap process[30,32,33]. An example Al/$AlO_x$/Al overlap JJ is visualized with a scanning electron microscopy (SEM) image in Fig. 1c, and a cross section of the JJ is detailed in a transmission electron microscopy image in Fig. 1d.

The fabrication process for the overlap JJ described in this work is an extension of the methodology described in our previous research[30,34], with the addition of all-optical lithography and the native incorporation of advanced metrology structures in the mask design for process control and monitoring, which resulted in better devices with more consistent behaviour. Fig. 1e provides a schematic representation of the critical steps in the process. In the initial stage (step I in Fig. 1e), the bottom electrode of the JJ was patterned alongside other circuit components, such as resonators and ground planes, in the first design layer. A 70 nm Al film was sputtered at room temperature onto a hydrofluoric-acid-cleaned, high-resistivity ($R_s \geq 3$ kΩ cm) Si substrate. The first design layer was patterned by optical immersion lithography (193 nm). After exposure, the pattern was transferred to the Al film by subtractive Cl-based dry-etching, which was followed by wet-cleaning with diluted sulfuric acid and peroxide and rinsing with deionized water. In the second stage (steps II–IV in Fig. 1e), the barrier formation process involved the complete removal of the native Al and Si oxide through argon milling, followed by regulated dynamic oxidation. This $AlO_x$ barrier was then overlaid in situ with a 50 nm Al film sputtered at room temperature. In the final stage (step V in Fig. 1e), the top electrode was patterned in the second design layer using an analogous process to the patterning of the bottom electrode with optical immersion lithography and dry-etching. Upon completion of fabrication, the wafer was coated with a protective resist layer before dicing. This resist was subsequently removed from the subdies using acetone, isopropanol and a brief 2 min oxygen ashing process. Finally, the subdies underwent a post-etch residue removal treatment (EKC) before they were wire-bonded into a measurement package.

## Qubit coherence

The described fabrication process was benchmarked in terms of qubit energy relaxation times across the wafer. 32 subdies D1 and 24 subdies D2 from across a single 300 mm wafer were measured at 10 mK in a dilution refrigerator (see our previous work[30] for set-up details) without being screened or pre-selected. Out of 400 qubits, 393 (yield 98.25%), with a variety of different designs and sizes, were functional and fully characterized (Supplementary Information). We estimated an upper

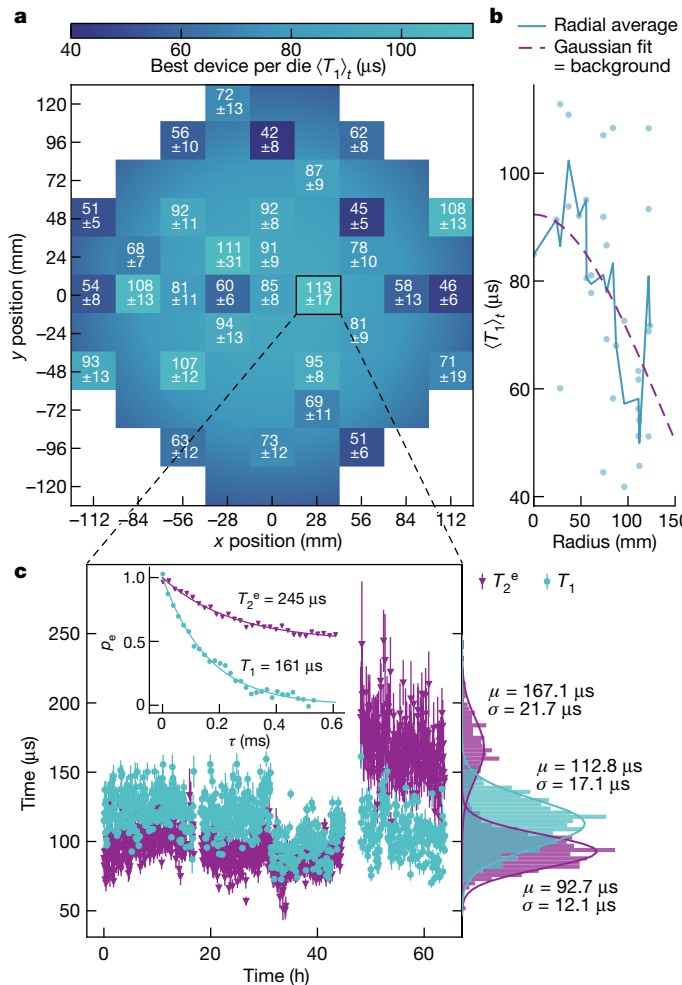

**Fig. 2 | Qubit relaxation and coherence times. a**, Wafer map of time-averaged relaxation times $\langle T_1 \rangle_t$ of the best-performing qubit on each measured die. The mean values and standard deviations are printed at each measured location. **b**, The background colour (**a**) represents a heuristic Gaussian fit to the average value as a function of radius. **c**, Repeated measurements of the relaxation time $T_1$ and Hahn-echo coherence time $T_2^e$ of the best-performing qubit on die (1,0) over 60 h. The time statistics are represented by histograms with Gaussian fits of mean $\mu$ and standard deviation $\sigma$. Inset, example traces of the relaxation and decoherence of the qubit excited state population ($p_e$) as a function of delay time ($\tau$).

bound on the qubit relaxation time for our current process by monitoring the time-averaged relaxation times $\langle T_1 \rangle_t$ of the best-performing qubits from subdies D1 across the wafer. A centre-to-edge dependence was observed. The highest value $\langle T_1 \rangle_t^{max} = 113$ µs was measured close to the centre. The lowest value $\langle T_1 \rangle_t^{min} = 42$ µs was measured near the edge. The median was $\langle T_1 \rangle_t^{med} = 75$ µs (see the wafer map in Fig. 2a). $\langle T_1 \rangle_t$ as a function of distance from the centre of the wafer is visualized with a background colour map in Fig. 2a, which illustrates the radial symmetry. A heuristic Gaussian fit represents the data well (Fig. 2b), with only a few outliers deviating from the trend. This centre-to-edge effect is commonly observed in most parameters controlled by wafer fabrication and indicates the importance of advanced process control. As modern semiconductor tooling allows, in principle, such levels of control, our results also indicate a clear road towards even better, batch-to-batch identical devices in the future. However, already on a per die level (which matters most for quantum processors), the present results show remarkable performance.

To ensure objectivity and prevent cherry-picking, all reported $T_1$ values were time-averaged (on average 20 h). An example time trace is shown in Fig. 2c. As commonly observed in transmon devices[35,36], we found significant temporal variation ($\sigma \approx 20$ µs) in both relaxation and coherence times with near Gaussian distributions reaching $T_1 = 161$ µs and $T_2^e = 245$ µs (inset in Fig. 2c). Such temporal fluctuations are associated with coupling of the qubit to near-resonance two-level system (TLS) defects[37], which are, in turn, longitudinally coupled to low-frequency (thermally active at 10 mK) two-level fluctuators (TLFs)[35,38,39]. The example trace in Fig. 2c exhibits a jump in the average Hahn-echo coherence time $T_2^e$ of six standard deviations after 45 h of measurements, without any known external trigger. The absence of a concurrent jump in $T_1$ indicates that the most plausible explanation is the vanishing of a coupled TLF (and not a near-resonant TLS), possibly due to a defect rearrangement from an impinging high-energy radiation event[40] or due to a trapped quasiparticle escaping from a shallow, local well in the superconducting gap[41].

## Coherence limitations and interface defects

Following the comprehensive analysis of qubit quality across the wafer, a deeper exploration of qubit relaxation and coherence limitations was warranted. The transmon qubits on subdies D1 were designed with four capacitor geometry sizes, resulting in a sixfold difference in the calculated participation ratios of the electric field over energy (EPRs) at qubit capacitor interfaces (metal–air, substrate–metal and substrate–air)[42]. This allowed us to differentiate between loss sources at capacitor interfaces from other losses, such as bulk substrate loss and more importantly losses induced by the JJ. The qubit energy loss ($1/Q$) scaled linearly with the sum of all interface EPRs (Fig. 3a, with measured averages $Q = \{1.7, 1.1, 0.84, 0.42\}$ million). A linear loss model fits the data well:

$$\frac{1}{Q} = \frac{1}{2\pi T_1 f_{qb}} = \delta_0 + \delta_{SA} p_{SA} + \delta_{SM} p_{SM} + \delta_{MA} p_{MA}$$
$$= \delta_0 + \delta_t (p_{SA} + p_{SM} + p_{MA}). \quad (1)$$

In equation (1), $\delta_0$ encompasses all the qubit relaxation channels that are not at capacitor interfaces. $p_{SA}$ is the substrate–air interface EPR, $p_{SM}$ the substrate–metal interface EPR and $p_{MA}$ the metal–air interface EPR, which are all proportional to each other[42]. $\delta_{SA}$, $\delta_{SM}$ and $\delta_{MA}$ are the relaxation losses at those respective interfaces, and $\delta_t$ represents the effective total interface loss. Projecting the total interface participation to zero yields $\delta_0 = (1.67 \pm 1.69) \times 10^{-7}$, comparable with previously reported values for similar Si substrates[43,44], although here it also includes JJ-induced losses. This corresponds to a $T_1$ limit of approximately $0.3_{0.15}^{\infty}$ ms at $f_{qb} = 3$ GHz, considerably greater than the mean values depicted in Fig. 2a, which affirms that, at present, qubit relaxation is predominantly dictated by capacitor interface losses. Interface engineering techniques with metal seed layers or metal encapsulation[45] may improve this process. Furthermore, our results imply that the fabrication of the overlap JJ, including the physical damage to the junction's bottom electrode due to argon milling, did not cause a noticeable rise in TLS defect losses within the Al/AlOₓ/Al junction. This is consistent with the previously observed resilience of Al to TLS defects induced by argon milling[34,46].

To gain insight into the coupled TLS defects, a magnetic-flux-tunable qubit on a subdie D2 was used to scan the spectral environment. The qubit frequency as a function of the applied magnetic flux did not exhibit any observable avoided crossings with the strongly coupled TLS (more than 250 kHz; Supplementary Information), reminiscent of a TLS inside the JJ barrier[47,48]. However, individual, weakly coupled, TLS defects were spectrally resolved with swap spectroscopy[39,49]. We excited the qubit with a π pulse and then left it to relax for a fixed duration while it was being detuned in frequency by a flux pulse, as shown by the pulse sequence in the inset of Fig. 3b. This procedure was used with different flux pulse amplitudes to scan a frequency range of 500 MHz repeatedly for 13 h.

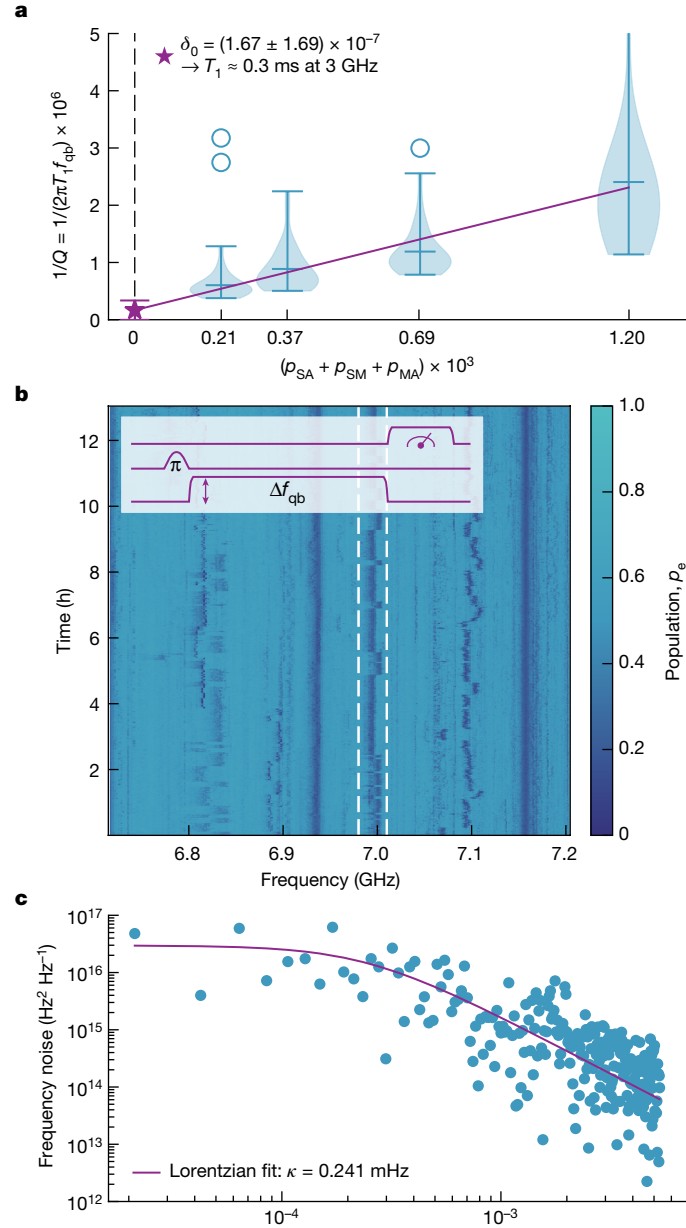

**Fig. 3 | Interface TLS defects. a**, Qubit loss ($1/Q$-factor) as a function of the total interface participation ratio (substrate–air, metal–air and substrate–metal) of the electric field energy for four different qubit designs on subdies D1 across the wafer. The distributions of each design were filtered for outliers beyond the interquartile range and visualized with violin plots including the mean and extrema (outliers shown as scatter points). The linear fit extrapolates the loss to zero interface participation and extracts the upper bound on the relaxation time. **b**, A flux-tunable qubit population as a function of qubit frequency measured repeatedly over 13 h to visualize the dynamics of relaxation channels associated with TLS defects at interfaces. Inset, the pulse sequence used to scan the spectrum. **c**, Frequency noise power spectral density of the highlighted defect in **b**, including Lorentzian fit with full-width at half-maximum linewidth $\kappa$.

The resulting time–frequency map of the residual qubit population, visualized by Fig. 3b, clearly shows several spectrally resolved relaxation channels. These relaxation channels are probably weakly coupled TLS defects at the capacitor interfaces, where the electric fields were orders of magnitude lower than inside the JJ[3,50]. Most of the defects showed random telegraph-noise-like switching, which we attributed to the coupling of these TLSs with a TLF[35,38,39]. We estimated the density of the TLSs to be approximately 34 GHz$^{-1}$ (Extended Data Fig. 1), which

is comparable to values reported for shadow-evaporated qubits[3]. One of the fluctuating TLSs was isolated and its frequency noise power spectral density is shown in Fig. 3c. The frequency noise power spectral density of the TLS was fitted well with a single random TLF model characterized by a Lorentzian with a TLF random switching rate ($\kappa$) of 0.241 mHz, comparable to timescales of previously reported relaxation time fluctuations[36]. The random jump in $T_2^e$ observed in Fig. 2c could, therefore, be explained by the disappearance of a similar TLF (with the switching rate faster than the inverse measurement time) and the consequential stabilization of the coupled TLS.

## Qubit frequency variability and ageing

The variability of qubit frequencies and JJ resistances were examined as indicators for the control and variability of the fabrication process. The variability in the transmon qubit frequency arose primarily from variations in the JJ critical current.

The Ambegaokar–Baratoff relation[51] for the critical current of a JJ describes the link between a transmon qubit's frequency[15] and the normal state resistance of the junction ($R_n$), $f_{qb} \propto \sqrt{1/R_n}$. Junction resistance is determined by the overlapping area ($A$) between the two junction electrodes, the barrier thickness ($d$) and barrier resistivity ($\rho$). Consequently, the variation of the qubit frequency is determined by the variations of these quantities. In the context of fabrication, the variation of the JJ area is primarily controlled by patterning processes, whereas the barrier oxidation processing step regulates the variability of both the barrier resistivity and thickness. It is, therefore, prudent to monitor the variability of the JJ area and the product of resistance and area, which encapsulates the resistivity and thickness into a single parameter RA that is representative of the overall oxidation uniformity.

Normal state resistances were collected for 7,872 JJ test structures across the wafer using a 300 mm wafer prober. Both the resistance and the relative standard deviation (RSD) scale with the estimated junction area (Methods and Fig. 4a,b). Approximately half of the variability can be attributed to an approximately 20% centre-to-edge decrease of the average $R_n$ (Supplementary Information). This was corroborated by an approximately 50% reduction in RSD$_{R_n}$ on a single die (pink data) compared to the across-wafer statistics (purple data) in Fig. 4a. Furthermore, the dependence of the $R_n$ RSD on the JJ area fits well with a model of constant area variance ($\sigma_A^2$), derived by propagating the uncertainty[3,52] (Methods):

$$\text{RSD}_{f_{qb}} = \frac{\text{RSD}_{R_n}}{2} \approx \frac{1}{2} \sqrt{\text{RSD}_{RA}^2 + \frac{\sigma_A^2}{A^2}}. \tag{2}$$

Fitting the data to equation (2) allowed us to disentangle the barrier non-uniformity and area variability. This analysis revealed RSD$_{RA}$ = 4.47% and $\sigma_A$ = 0.00334 μm$^2$ on a single die, meaning that, for all JJ with $A > 0.075$ μm$^2$, barrier non-uniformity was the dominant cause of $f_{qb}$ variability. The standard deviations for area are comparable with values reported in literature[3,53].

The RSD of the measured qubit frequencies on subdies D2 across the wafer (excluding the magnetic-flux-tunable qubit) compares well with the expected relation RSD$_{R_n}$ = 2RSD$_{f_{qb}}$ (following $f_{qb} \propto \sqrt{1/R_n}$) (Fig. 4a). The RSD$_{f_{qb}}$ ranged between 5% and 7% for the different junction areas tested, with no discernible trend with area. Across the wafer, locally averaged qubit frequencies (including the JJ area correction $f_{qb}/\sqrt{A} \propto 1/\sqrt{RA}$) exhibited an approximately 10% centre-to-edge increase (Fig. 4c), in agreement with JJ test array $R_n$ wafer maps (Supplementary Information).

Leveraging the possibility of even more advanced process control, as available with modern tooling, we can envisage ways to further push the homogeneity of qubit frequency towards consistent across-wafer, die-to-die values. Our analysis indicates that significant variability was due to the controlled oxidation process, which can be further

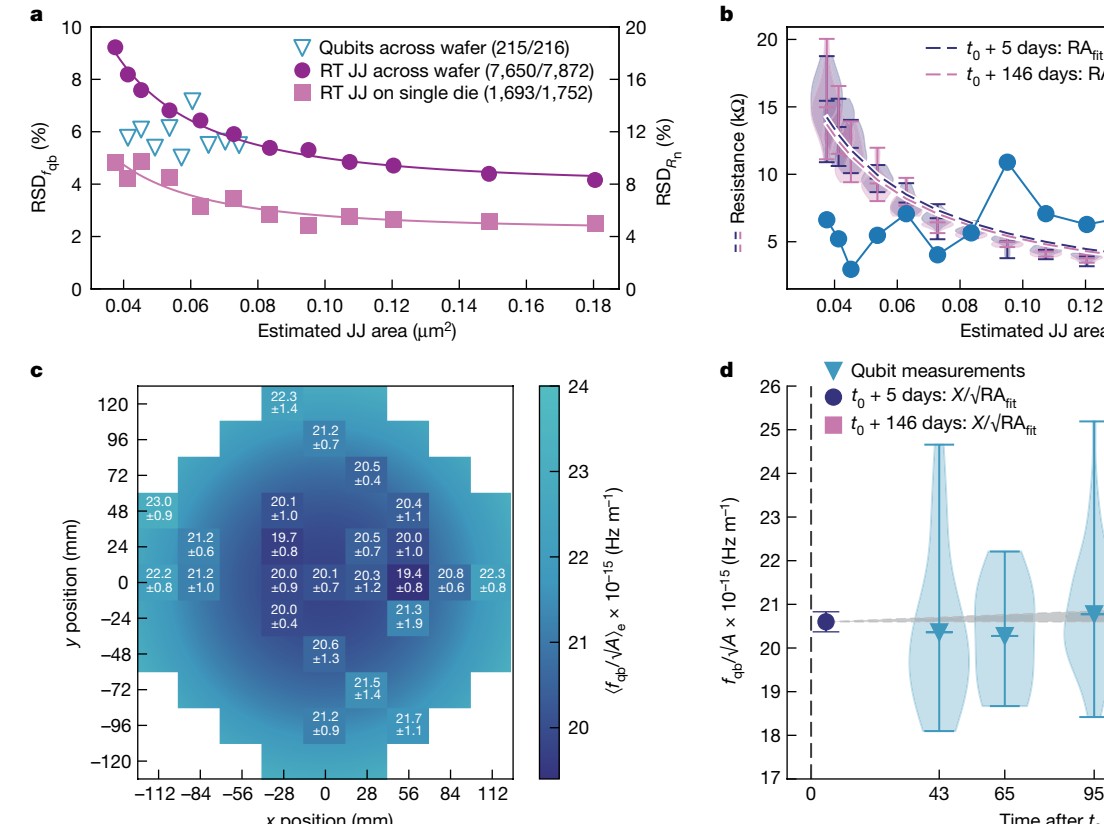

**Fig. 4 | Qubit frequency variability and ageing analysis. a**, RSDs of qubit frequencies and normal state resistances of JJ test structures ($\mathrm{RSD}_{f_{qb}} = \mathrm{RSD}_{R_n}/2$, visualized with the double $y$ axes) as a function of the estimated JJ area. 9 (qubits per die) × 24 (dies) = 216 qubits across the wafer, 8 (JJs per area per die) × 12 (areas) × 82 (dies) = 7,872 JJs across a wafer, 146 (JJs per area) × 12 (areas) = 1,752 JJs on a die (0,2). All working qubits are included. The resistances of JJ test structures were filtered per area for outliers beyond 1.5 times the interquartile range. Solid lines are fits to equation (2). **b**, Normal resistances of JJ test structures as a function of estimated junction area for a subgroup of 216 devices measured 5 days after fabrication ($t_0$) and once more 146 days after fabrication. The right $y$ axis shows the corresponding change in the relative average resistance for each JJ area. **c**, Wafer map of qubit frequencies scaled

with $\sqrt{A}$ (where $A$ is the estimated JJ area) and ensemble averaged ($\langle\rangle_e$) over the nine qubits on each subdie D2. The mean values are accompanied by the standard deviation. The background colour represents a heuristic Gaussian fit to the average value as a function of radius. **d**, Area-scaled qubit frequencies of qubits on subdies D2 plotted as a function of cool-down time after fabrication, compared with the product of the resistance and area of the JJs extracted from the fits in **b** scaled with a proportionality factor $X \approx \sqrt{\Delta E_C/he^2}$ (where $\Delta$ is the superconducting gap of aluminium, $E_C$ is the qubit charging energy, $h$ is Planck's constant and $e$ is the elementary charge). The black vertical dashed line represents the wafer fabrication date at $t_0$. Distributions of measured qubits are represented by violin plots showing means and extrema. The grey area indicates the ageing from JJ data.

fine-tuned. The residual area variance could be addressed with more advanced lithographic control tricks and etch optimization beyond the range achievable at present with angled shadow-evaporation techniques, in principle[3,54]. Furthermore, higher-order contributions to the JJ resistance variation, for example, aluminium grain-size effects and the sidewall edge slope (which is impacted by etching and argon milling), could be addressed. Finally, at the design level, larger junction areas (with higher RA values) could be targeted to minimize the junction size variation, consistent with recent literature reports[3].

Shadow-evaporated JJs often show notable parameter ageing (an increase in the normal resistance as a function of time), with reported values as high as tens of percents, over periods spanning days to weeks[3,55,56]. The overlap junctions fabricated in this work, however, showed significantly less ageing, being limited to a decrease of 3.7% over 146 days (Fig. 4b). A subset of 216 JJ test structures was measured 5 days after fabrication, and once more 146 days following fabrication (the diced wafer was stored covered by a protective resist in a clean-room environment at room temperature). Furthermore, junction ageing can also be tracked by comparing $f_{qb}/\sqrt{A}$ ($\propto 1/\sqrt{RA}$) of subdies D2 as a function of their cool-down date over 151 days (Fig. 4d). The aged RA values, obtained from measurements of the JJ test structures (rescaled to match the qubit values) illustrate that any detected ageing falls within the

measured qubit frequency variation. Ageing can be linked to barrier impurities[57] or oxygen diffusion[3,58]. In the presented fabrication process for overlap JJs, no organic material was present during the formation of the barrier, in contrast to the conventional fabrication process where an organic photoresist is used as the angled shadow mask. We hypothesize, therefore, that the weak ageing effects in the JJ resistance of the presented process could potentially be attributed to the absence of carbon impurities within the barrier (Supplementary Information), though further studies would be required to substantiate that hypothesis.

## Conclusion and outlook

In summary, our study presents a large-scale fabrication process for superconducting qubits using fully industrial semiconductor nanofabrication methods on 300 mm silicon wafers, achieving high coherence and an across-wafer yield of 98.25% in an industry-standard facility. The process quality was confirmed through large-scale statistics of qubit relaxation and coherence measurements conducted across the wafer. The time-averaged times $T_1$ and $T_2^e$ exceeded 100 µs. Our measurements shed light on observed centre-to-edge dependencies and suggest an avenue for further improvements by leveraging the advanced process control of modern semiconductor nanofabrication tooling.

Our investigation identified the dominant relaxation and decoherence sources as TLS defects at capacitor interfaces. Our variability analysis of the JJ normal resistance and qubit frequency indicated that the current limitation resides in barrier oxidation. However, the area control realized through optical lithography and subtractive dry-etching shows clear promise for further optimizations beyond the state of the art. Finally, we verified the stability of our process over at least 151 days. This underscores that the fabrication process is robust and reliable.

Looking ahead, the qubit fabrication process presented in this work, aligned with CMOS foundry standards, establishes a robust baseline for future enhancements. The integration of industry-standard techniques (such as chemical mechanical polishing, silicon trenching and the deposition of sophisticated material stacks) alongside three-dimensional integration, will provide the necessary yield, uniformity and connectivity needed to fabricate increasingly complex quantum processors.

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

## Methods

### TLS density estimation

The TLS density was estimated from the time–frequency map in Fig. 3b. The data were first sliced along the time axis into frequency traces, as exemplified in Extended Data Fig. 1. The number of TLSs in each slice was counted using the Python scipy.signal.find_peaks() method. The number of counted TLSs was then averaged over all timestamps, resulting in a density of 34.4 TLSs per gigahertz. Note that the number of identified TLSs and, therefore, the extracted density depend on the settings of the peak-finding algorithm. In the analysis, the prominence was set to 0.15.

### Estimating the JJ area

The actual JJ areas deviated significantly from the designed values. A challenging aspect of dimension targeting arises from the tapering of the bottom-electrode sidewalls (followed by argon milling), which is necessary for good step coverage in the deposition of the top electrode. The process flow presented in this work could be further optimized by, for example, fine-tuning the etching process and making optical proximity corrections to improve targeting of the critical dimension. The JJ analysis described utilizes a 'best-effort' area estimate for the JJ. First, an offset on the designed critical dimensions (the widths of the bottom and top electrodes in the junction) was determined from SEM images taken across the wafer, as illustrated in Extended Data Fig. 2a,b. A constant correction of the critical dimensions of the bottom and top electrodes was extracted from a fit to the SEM inspections (Extended Data Fig. 2d). Second, the rounded surface of the bottom electrode, which was due to the argon milling, was accounted for with an approximate formula for the circumference of an ellipse. The major radius of the ellipse was gathered from the bottom-electrode critical dimension, whereas the minor radius was extracted from the transmission electron microscopy image in Extended Data Fig. 2c ($h_{BE} = 59$ nm). Extended Data Fig. 2e illustrates the best-effort area estimates for each designed nominal critical dimension. The best-effort approach allowed us to fit the JJ normal resistance as a function of area (Fig. 4b) with only one free fitting parameter RA and no need for any further area corrections or offset series resistance (besides the included resistance of 32 Ω measured on shorted test structures to account for probe contact resistance). Note that any location dependence of the area, although observed (Supplementary Information), was not included in the area estimates as there were insufficient data for all locations. The observed centre-to-edge drift is discussed in more detail in the Supplementary Information.

### RSD analysis of JJ resistance

The RSD of the JJ normal resistance shows a functional dependence on the JJ area, as shown in Fig. 4a. Modelling this area dependence can be leveraged to disentangle variance contributions of area and barrier uniformity (tracked with the product of resistance and area RA). Propagation of uncertainty was used for $R_n = RA/A$ to arrive at $RSD_{R_n} \approx \sqrt{RSD_{RA}^2 + RSD_A^2}$, for which a logical assumption was that the barrier non-uniformity was independent of JJ area. The observed area dependence of $RSD_{R_n}$ was then entirely attributed to $RSD_A = f(A)$. The area of the JJ fabricated in this work includes the rounded surface of the bottom electrode and is described by the formula in the legend of Extended Data Fig. 2e. The total area variance $\sigma_A$ of these junctions have contributions from variations in the bottom-electrode critical dimension ($cd_{BE}$), variations in the top-electrode critical dimension ($cd_{TE}$) and variations of the bottom-electrode thickness ($h_{BE}$). At present, we do not have sufficient data to model the area variance in its entire functional dependence $\sigma_A = f'(cd_{BE}, cd_{TE}, h_{BE}, A)$, so we considered two models. In model A, we assumed that the area variance was independent of area, $d\sigma_A/dA = 0$, such that $RSD_A = \sigma_A/A$. In model B, we oversimplified the area calculation as $A = cd^2$, with $cd = cd_{BE} = cd_{TE}$ and $d\sigma_{cd}/dA = 0$, such that $RSD_A = 2\sigma_{cd}/\sqrt{A}$, as in the literature[3,52].

Extended Data Fig. 3a shows fits to both models of the measured JJ normal resistance RSD data (across wafer and on one single die) in Fig. 4a. The best-fitting parameter values are reported in Extended Data Table 1. Both models produced fitted curves that represent the data well. However, the extracted fitting uncertainties of model A were considerably lower than those of model B (Extended Data Table 1). Therefore, we proceeded with our analysis using model A. Importantly, according to model A, the contribution of the barrier non-uniformity to the measured resistance variability is dominant for a JJ with $A > 0.075$ $\mu m^2$, whereas according to model B, all measured structures are dominated by area variability. This is a conclusion drawn from Extended Data Fig. 3b, as we calculated and compared $RSD_A$ for both models with data collected from a single die.

According to the model A analysis, we conclude that the barrier oxidation uniformity contributed significantly to the reported $RSD_{R_n}$ values, as corroborated by the plateau in the RSD at large JJ areas in Fig. 4a (approximately 8% at wafer level and approximately 4% at the single-die level). We know from trilayer junctions (Nb/Al/AlO$_x$/Nb) with similarly grown AlO$_x$ barriers at our facility that $RSD_{R_n} < 0.8\%$ for large area junctions are attainable[26] (an upper bound for the RA variability). We are confident that by optimizing the dimension control (optical proximity effect corrections, resist stacks and etching conditions) and the bottom-electrode morphology control (edge slopes, grain sizes and surface roughness), we can significantly reduce the area variability. With optimized 193 nm immersion lithography, subnanometre standard deviations on the critical dimension of Si etched trenches are possible, and more advanced technology nodes (extreme ultraviolet and high-numerical aperture extreme ultraviolet) could potentially push this even lower. We believe an optimized aluminium process should eventually reach similar values. We are hopeful that we can ultimately reach an $RSD_{R_n}$ for reasonable junction sizes of below 1.0%, which is within the required range of 0.5–1.0% resistance variability targeted by fixed-frequency qubit processor architectures[59], without the need for laser annealing.

### Interface participation ratio calculation

The different qubit designs on subdie D1 have planar capacitors resembling a section of a coplanar waveguide (Supplementary Information), which simplified the analytical calculation of the electric field versus energy participation at the interfaces[42] of the qubit capacitor. The substrate–air, metal–air and substrate–metal interface participation ratios were calculated for the four different capacitor geometries (width and gap in {13, 24, 48, 90} μm). We used interface thicknesses of 5 nm for the metal–air and substrate–metal interfaces and 3 nm for the substrate–air interface. The relative dielectric constants used in the calculations are $\epsilon_{sub} = 11.9$ (for the Si substrate), $\epsilon_{MA} = 10$ (for the Al oxide), $\epsilon_{SM} = 11.9$ (for the Si–Al interface) and $\epsilon_{SA} = 3.9$ (for the SiO$_2$). The calculated interface participation ratios for the four different qubit geometries are plotted in Extended Data Fig. 4. The corresponding substrate participation ratios were {91.8%, 92.0%, 92.1%, 92.1%}, and the vacuum participation ratios were {7.93%, 7.86%, 7.82%, 7.79%}.

The calculated participations of the substrate–air, metal–air and substrate–metal interfaces are proportional to each other, which allowed us to group them into a total interface participation of our simple loss model in equation (1). Note that the analysis and conclusions in the text do not rely on the exact values of the calculated participation ratios, but only on scaling the interface contribution with respect to device geometries. Under the assumptions made for the interface thicknesses and dielectric constants, the effective interface loss tangent of our devices is $\delta_t = (1.79 \pm 0.23) \times 10^{-3}$, which is slightly better than the equivalently calculated values of $\delta_t = (p_{MA}\delta_{MA} + p_{SA}\delta_{SA} + p_{SM}\delta_{SM})/(p_{MA} + p_{SA} + p_{SM}) \in [2.23; 18] \times 10^{-3}$ for previously reported aluminium on silicon resonators[44].

## Data availability

The data supporting the findings of this study are available at Zenodo (https://zenodo.org/records/13143313)[60].

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

**Acknowledgements** We are grateful for the support of the P-line of Imec, operational support, the Materials and Components Analysis (MCA) team, K. Verhemeldonck for wire-bonding support and F. Loosen for taking pictures. A. Mahbub is acknowledged for supporting the JJ resistance analysis. This work was supported in part by the Imec industrial affiliation programme on quantum computing. J.V.D. acknowledges the support of the Research Foundation-Flanders through the Strategic Basic Research PhD programme (grant no. 1S15722N). We thank O. Painter, J. Bylander, C. Haffner and P. McMahon for insightful comments on this work.

**Author contributions** S.M. coordinated the fabrication process, with etch development by Y.C., thin-film development by D.P.L., lithography development by D.V. and Y.H., and clean development by J.G.L. Ts.I. developed the post-processing and sample preparation procedure. A.P., M.M. and J.V.D. designed the qubit samples. J.V.D., R.A., M.D. and A.P. performed the measurements and analysed the qubit data at cryogenic temperatures. S.M. and Ts.I. measured the JJ test arrays at room temperature. J.V.D. analysed the gathered JJ test array data. J.V.D., R.A., A.P., M.M. and A.M.V. prepared the experimental set-up and methods. J.V.D. prepared the manuscript, with input from all authors. A.P., M.M., D.W., J.D.B. and K.D.G. supervised and coordinated the project.

**Competing interests** The authors declare no competing interests.

**Additional information**
**Correspondence and requests for materials** should be addressed to A. Potočnik.

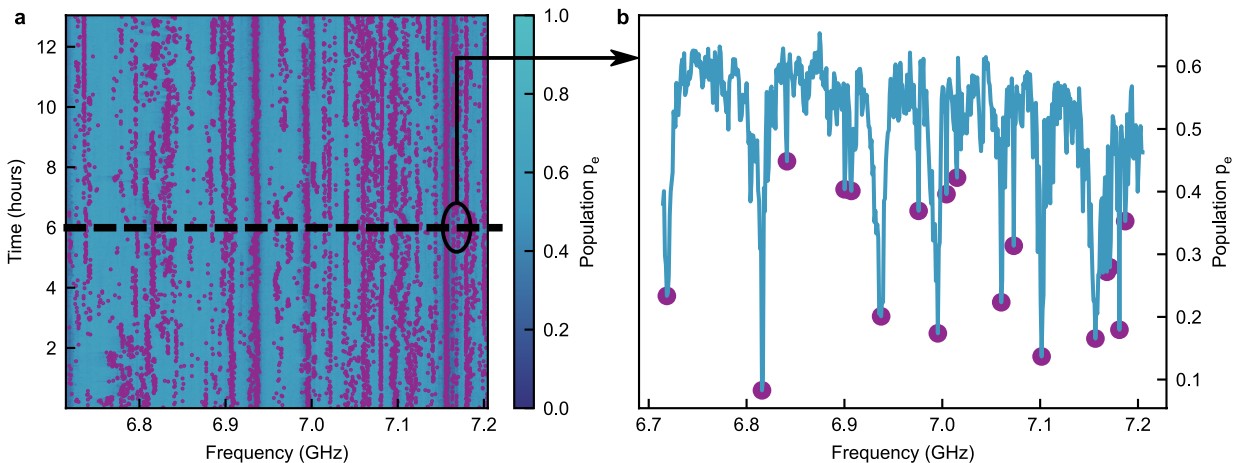

**Extended Data Fig. 1 | Two-level system counting. a**, Time-frequency map (main text Fig. 3b) with highlighted two-level systems. **b**, Example slice of **a** illustrating the detected TLS at one timestamp.

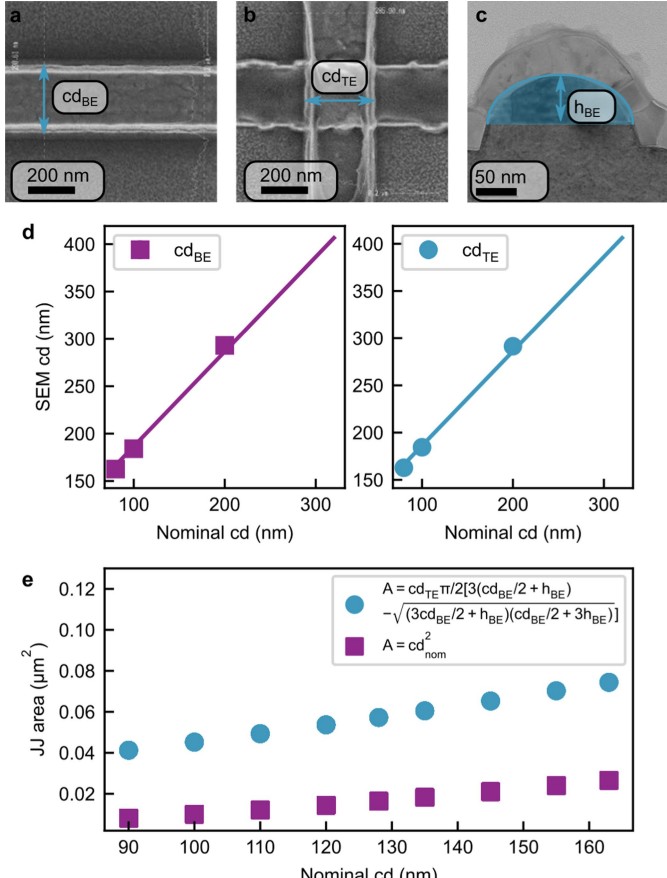

**Extended Data Fig. 2 | JJ area estimation. a**, SEM image of a JJ after BE patterning. **b**, SEM image of a JJ after TE patterning. **c**, TEM image of a cross section of a JJ, including an ellipse circumference as approximation of the rounded BE surface. **d**, Average width of the JJ's BE and TE measured from the SEM images **a**,**b**. The solid lines are linear fits including a constant offset. **e**, Best effort estimate of the JJ area, calculated for each designed JJ critical dimension.

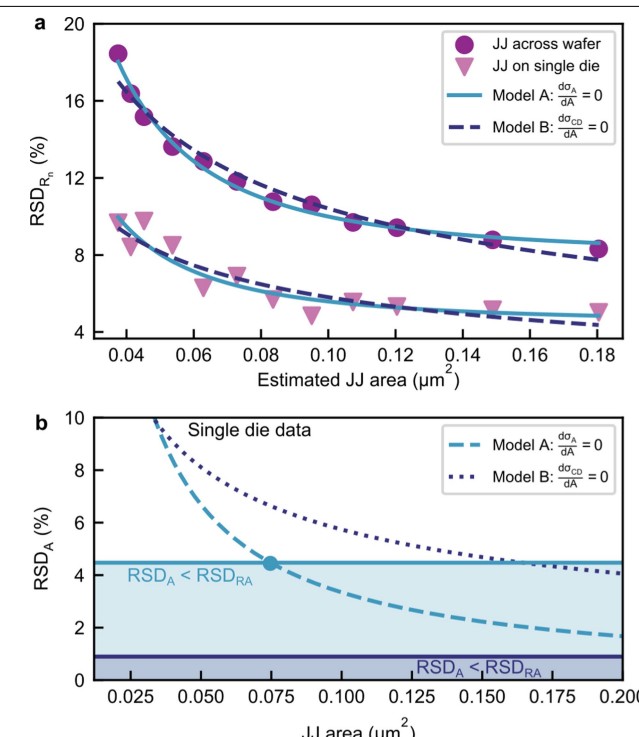

**Extended Data Fig. 3 | Resistance variance area dependence model comparison. a**, Relative standard deviation of JJ normal resistances measured across the wafer and on a single die as function of the estimated JJ area (same data as Fig. 4a). The solid lines represent the best fit with model A of constant area variance ($d\sigma_A/dA = 0$), the dashed line is the best fit with model B of constant critical dimension variance ($d\sigma_{cd}/dA = 0$). The corresponding fit parameter values are presented in Extended Data Table 1. **b**, The relative standard deviation of the JJ area as function of area is calculated for both models from the best fit values in Extended Data Table 1 (only single die data shown for figure visibility). The horizontal lines highlight the cross-over from area variance dominated ($RSD_A > RSD_{RA}$) to barrier uniformity limited ($RSD_{RA} > RSD_A$) RSD of the JJ normal resistance for model A and model B, respectively.

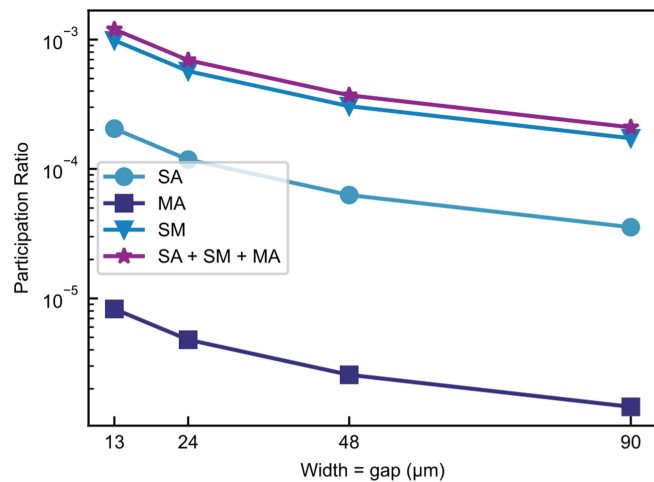

**Extended Data Fig. 4 | Interface participation ratios.** The calculated electric field energy participation ratios calculated for the four different qubit capacitor geometries on sub-die D1.

## Extended Data Table 1 | Resistance variance model parameters

| | | RSD$_{RA}$ (%) | $\sigma_A$ ($\mu$m$^2$) | $\sigma_{cd}$ (nm) |
|---|---|---|---|---|
| Model A | across wafer | 7.94 $\pm$ 0.18 | 0.00608 $\pm$ 9.9 $\times$ 10$^{-5}$ | |
| | single die | 4.47 $\pm$ 0.37 | 0.00334 $\pm$ 2.0 $\times$ 10$^{-4}$ | |
| Model B | across wafer | 5.87 $\times$ 10$^{-6}$ $\pm$ 6944 | | 16.5 $\pm$ 0.51 |
| | single die | 0.895 $\pm$ 3.2 | | 9.07 $\pm$ 0.62 |

Best fit parameter values of models A and B fitted to the data in Extended Data Fig. 3. The values are reported together with the standard error in fit uncertainty.