## [Peer Review File · Nature]

Manuscript Title: Advanced CMOS manufacturing of superconducting qubits on 300 mm wafers

Reviewer Comments & Author Rebuttals

Reviewer Reports on the Initial Version:

Referees' comments:

Referee #1 (Remarks to the Author):

The authors of this manuscript describe their work on fabricating superconducting qubits using semiconductor CMOS fabrication technology on 300 mm Si wafers. They report statistics of the qubit relaxation and coherence times across the wafer. They conclude that their approach, based solely on optical lithography and removal of materials by reactive ion etching, promises a better way of upscaling of superconducting quantum processors when compared to widely adapted metal lift-off, multi-angle evaporation (such as Dolan bridge and Manhattan techniques), and electron-beam writing.

This work is a continuation of the authors' work published in npj Quantum Information in 2022. [1] The fabrication techniques, characterization of their qubits, and even the structure of the manuscript follows closely their earlier work. To be fair, the authors carried an extensive study to include characterization of hundreds of qubits as well as thousands of Josephson junctions. However, in my opinion, this work does not rise to the level of a Nature publication. My conclusion is based on the followings:

- Scaling up the number of qubits is certainly important for quantum information processing. However, maintaining high qubit coherence and relaxation times while scaling up is just as important. The recent development in the field is pushing average T1 times beyond ~ 300 us, several times better than what is achieved in this work. [2-4]
- The relaxation times and Hahn-echo times reported in this work are for isolated qubits. It would be interesting to see how these times are holding when these qubits are coupled to each other. Are the authors planning experiments on coupled qubits?
- There is significant variation of performance between qubits in the center and near the edge of the wafer. No clear explanation of this discrepancy is offered. Is it the variation in dry etching, or Ar milling, or sputtering? Could the authors design experiments to point which processing steps is the likely cause?
- In my opinion, Eq. 1 is simplified without a good justification. Why did they have to use an effective total interface loss δ_t ? How do the individual losses compare to each other at the SA (silicon oxide), SM (some kind of silicide and/or Si oxide), and MA (aluminum oxide)? δ_0 are primarily due to Si substrate bulk losses. How does it compare to published literature?
- The authors state in the manuscript "The qubit frequency as function of applied magnetic flux did not exhibit any observable avoided crossings with strongly coupled (> 250 kHz) TLS (data not shown)". Why not show the data in the supplementary section?
- The authors state in the manuscript:

“Our analysis indicates that significant variability is currently coming from the controlled oxidation process, which can still be further fine-tuned. Residual area variance can be addressed with more advanced lithographic control tricks, and etch fine tuning, in principle beyond the range currently achievable with angled shadow evaporation techniques^{3,51}. Furthermore, higher order contributions to the JJ resistance variation e.g. aluminum grain-size effects, and sidewall-edge-slope (impacted by etch and argon milling) could be further addressed. Finally, at design level, larger junction areas (with higher RA values) can be targeted, minimizing junction size variation, consistent with recent literature reports³.”

The authors are discussing potential engineering controls to limit variation across the wafer. Similar engineering controls can be implemented with angled shadow evaporation techniques. I don't see how their technique is better in this respect.

- The authors show significant variation in JJ resistance from batch-to-batch and wafer-to-wafer within the same batch, especially for the smallest JJs. Do the authors understand why they observe such a big variation? and how they plan to address this?

- The authors state in the manuscript:

“Shadow evaporated JJ often show notable parameter aging (increase in the normal resistance as a function of time) with reported values as high as tens of percents, over periods spanning days to weeks^{3,52}. Overlap junctions fabricated in this work, however, show significantly less aging, limited to a decrease of 3.8% over 146 days”

Again aging in Shadow evaporated JJs could be addressed in the fabrication by introducing a gentle O₂ descumming step. [5]

- The authors focus on qubits based on Al. As discussed before, there are other material platforms promising better qubit coherence times. Could they incorporate these emerging materials into their processing? If so, how? There is also recent work on Nb trilayer junction qubits also based on all optical lithography. [6] With much larger superconducting gap compared to Al, Nb qubits could be less prone to quasiparticle loss. Could the authors comment on how Nb trilayer JJ qubits compares to their qubits?

[1] npj Quantum Information (2022) 8:93

[2] Nature Communications volume 12, Article number: 1779 (2021)

[3] npj Quantum Information volume 8, Article number: 3 (2022)

[4] arXiv:2304.13257

[5] J. Vac. Sci. Technol. B 30, 010607 (2012)

[6] arXiv:2306.05883

Referee #2 (Remarks to the Author):

Report on High-coherence superconducting qubits made using industry-standard, advanced semiconductor manufacturing, Van Damme et al.

Van Damme et al. demonstrate the fabrication of superconducting aluminum transmons on 300-mm wafers using essentially a CMOS line and toolset. The resulting Josephson junctions are tested at room temperature and show 5-8% cross-wafer variation, with high-90's hard yield (works versus doesn't work). The qubits exhibit coherence times in the 100-200 us range for the center of the wafer, reducing to 40 us towards the edge in a usual "donut" shape. The authors characterize and observe effects seen in the qubit community, including fluctuating TLS loss at the capacitor.

The notable accomplishments are:

1) Demonstrating reasonable transmon qubits (coherence and yield) in a 300-mm line using 193-nm optical immersion lithography, instead of the usual E-beam lithography.

2) Use of a "trilayer-type" Josephson junction formed with subtractive etching in multiple steps, instead of the usual single-step additive Dolan-style or Manhattan-style junction with lift-off.

3) Since the additive JJ approach does not have an exposed edge, the observed aging effect is much less than is typically seen with Dolan or Manhattan JJs, which generally have exposed edges.

4) Process monitoring of the junctions and qubits across a wafer with statistical characterization both at room temperature and a cryogenic temperatures.

I think this is wonderful to see, as having a commercially accessible foundry be able to produce high-quality and relatively high-yield superconducting devices is going to be a big advantage to academic groups (if they can afford it) and companies worldwide. As the authors say, the next steps would be to demonstrate 3D integration and heterogeneous packaging technologies that are also relatively mature in such foundries.

I very much appreciate the hard work that went into demonstrating the above results and congratulate the team for doing it. It is impressive to see several complex steps come together. Whether this is worthy of Nature is an editorial decision. In the spirit of reviewing the manuscript, here are a few "blue-team" and "red team" comments for the editors and authors to consider:

On the "blue team" side, in terms of novelty, I would say that the main innovation is the dome-shaped "trilayer" junction, which features:

- fabrication using multiple steps of subtractive etching. This can be contrasted with the usual Dolan-bridge or Manhattan-style approaches, which are generally additive approaches that use lift-off processing and generally a single deposition and oxidation step. Lift-off is known to be less reproducible and scalable than a subtractive etch approach.

- High coherence times, despite fabricating the junction in several steps. The bottom electrode is patterned and etched. Then, the metal surface is cleaned in an Ar plasma, and the clean surface oxidized. And then the top electrode is deposited and etched.

- The "dome" shape isolates and protects the junction oxide from further oxidation (to some

degree). This is likely one reason these devices age much less than conventional superconducting qubits. Note that previous Josephson devices made from niobium trilayers, both qubits and traveling wave parametric amplifiers, similarly do not age due to an anodization step that surrounds and protects the junction barrier. However, this adds an additional lossy oxide that tends to impact the qubit coherence. That additional anodization step is not needed with a “dome” junction. On the other hand, the lateral sides of the dome are still exposed to some degree, similar to a Manhattan junction. Late-stage static, dynamic, or plasma oxidation is sometimes used to “seal” the edges with AlO_x, and that may also be at play here. Whatever the reasons, the dome junction is an important innovation (but, see also below).

Mixed “blue-red”:

- However, the “dome” junction adds morphology to the fabrication. It requires precise control of a rounded dome that supports the junction barrier, for example, as well as control of the etching in to the silicon substrate. Congratulations to the team for getting it to work at close to the 99% yield level. However, we know that such morphology can be problematic when trying to achieve yields with more 9’s. This is one reason for the advent of planarization and planarized processing in the CMOS industry, particularly as the number of layers and complexity increases. This is something that the IMEC team will need to address as they go forward. Presumably, the motivation for 300-mm fab is to fabricate chips that contain thousands to millions of qubits, and so the yield needs to be at the six 9’s level (or more). And, at those numbers of qubits, more wiring layers and integration will be required. It will be interesting to see if the dome is compatible with such high yields and complexity.

On the “red team” side (and with the caveat that this work is likely only the first step of many to be taken):

- Most, if not all, of the processing steps have been demonstrated before in places like IBM, Lincoln Laboratory, and elsewhere, primarily on 200-mm wafers, although generally not published in this level of detail for various reasons (IP, trade secrets, etc.). I do not see a big difference between 200-mm and 300-mm wafers. The toolset is basically the same, and the advances made in the newer 300-mm tool technology are generally available in 200-mm bridge tools.

- I have seen public talks of high-coherence qubits using 193-nm lithography on 300-mm wafers from Albany NanoTech pre-COVID, and I am aware of unpublished works at other facilities. While 193-nm lithography is certainly faster (more manufacturable) than E-beam lithography, it is also much more expensive to implement (mask cost primarily) and slower to iterate (re-spinning masks). Nonetheless, I am sure that IMEC could similarly use E-beam lithography as well in this process.

- I am not aware of anyone in the above point using 193-nm *immersion* lithography. The immersion is used to obtain finer feature sizes, typically at the CMOS 65-nm node and below. I’m glad to see that the team has done this. However, despite the added fab complexity, it apparently did not improve the junction uniformity over other published results. This may not be too surprising, since much of the critical-current non-uniformity is speculated to arise from non-uniformity in the junction barrier.

- Almost all standard materials used in superconducting qubits are CMOS compatible, with perhaps the sole exception being gold pads that some groups use, which is easily replaced with other high-

conductance metals like platinum. I don't see this as a big differentiator. The main differences encountered in a CMOS foundry is that the temperatures generally used in CMOS fabrication are higher than would generally be allowed for superconducting qubit (JJ) fabrication. It does take process development to figure these steps out, e.g., how to bake the resist uniformly at lower temperatures, but that is technical process development.

Referee #3 (Remarks to the Author):

The manuscript reports the fabrication of superconducting qubits (transmons) with relatively high coherence (best T1s are more than 100us) in their own 300 mm advanced CMOS manufacturing facility. The reported statistics on yield, variability, and aging behaviour is very informative, particularly for researchers with limited access to advanced facilities like this and (understandably) little control over the manufacturing steps. The technical achievement reported in this manuscript is also very timely, the superconducting hardware community is currently pushing for devices beyond >100 qubits, and being able to harness some of the advantages of large-scale manufacturing while achieving high coherence will certainly be useful. There are numerous challenges that have to be overcome and the authors have done an impressive work surpassing them. As far as I know from published literatures, the reported T1 level is the first from such a facility, and hence will set the bar to surpass for the rest of advanced 300 mm manufacturing facilities.

The authors seem to have analysed their results in great detail, and present some of these analyses in the main manuscript. However I don't think that the presentation has been done in a way that is clear to the reader, and the majority of my feedback is related to this. I can recommend a publication in Nature after the issues/questions below have been addressed in a satisfactory manner (for me, and the readers in general).

(1) The authors claim to “*demonstrate the real advantages of advanced CMOS fabrication methods in terms of processing splits, process parameter control and advanced metrology to identify critical process steps*“. I have difficulty convincing myself that this is already described in the manuscript.

- Which parts of the manuscript show that they have made use of the process parameter control and advanced metrology to identify critical process steps?

- Furthermore, it looks like it is precisely the opposite that happened: the authors seems to have observed radial dependence in the qubit best T1s (figure 2a) as well as the frequency variation (figure 4c). I would have hoped that industrial manufacturing facility has better control over their processes, which in principle means that variations like this should not have happened in the first place?

(2) The authors choose to quote the “time averaged energy relaxation times ($\langle T_1 \rangle_t$) of best performing qubits from sub-dies D1“. I have difficulty in understanding this choice.

- Why are only best performing qubits relevant in this discussion as opposed to the overall T1 performance of all measured qubits? In the context of this manuscript, given the number of dies that can be produced, it is reasonable to expect that a substantial part of the measured T1 values are also reasonably good and that is perhaps what the authors should also describe to the reader.

- The notation $\langle T_1 \rangle_t^{\max}$ and $\langle T_1 \rangle_t^{\min}$ can be confusing. First of all, the specifiers “max” “min” are only referring to the fact that the $\langle T_1 \rangle_t$ values close to the center of the wafer are the highest and the ones close to the edge are the lowest? Can I instead suggest “center” and “edge”? Am I correct to state that these are measured values and not obtained from the interpolation?

- The 2D colormap plot, I have to admit, is not very clear. How do typical interpolated curves look when they are plotted in a 1D lineplot?

- Do the authors observe identical or similar center-to-edge dependence when considering the average T1 per die or the lowest T1 per die?

- It will be helpful for the reader to see the histogram of T1, T2* (or T2E) of all qubits that are measured in this wafer. The idea is to inform the readers how many qubits (out of 394) are actually crossing the 100us threshold. I appreciate the amount of work that has gone into this manuscript, I understand the challenges, and I am not looking to criticise the work if the majority of the T1 values are not above 100us.

- Finally, what are the frequency range of these qubits? Figure 3a shows the inverse of the qubit quality factor, can the authors states the quality factor as well in the text. This will be helpful as a quick reference for the reader.

(3) In figure 3a, the authors plot the inverse of the measured quality factor as a function of the simulated electric-field participation ratio of the interface (let's denote it as $p_{\text{interface}}$), fit it with a linear curve, extrapolate the expected quality factor when $p_{\text{interface}}$ becomes zero, and claim that the inverse of this extrapolated value represents that the T1 limit (around 300 us) that could be achieved if the $p_{\text{interface}}$ is actually zero.

- Am I correct in understanding the reasoning here?

- Am I also correct in stating that the linear relation is only valid if we assume that the participation ratio in the bulk volume (p_{bulk}) is unaffected when $p_{\text{interface}}$ goes to zero? I'm not entirely sure in which situation this ($p_{\text{interface}}$ goes to zero) would be possible. Even if we assume that it is possible, p_{bulk} should be affected as well and it is not straightforward to state that the linear relation remains valid for zero $p_{\text{interface}}$. Thus, the 300us limit may not mean much. I invite the authors to explain this part.

(4) Suppose that, for the moment, we agree that the 300us value is a meaningful limit.

- Do the authors have an idea of what is limiting the performance at the 300us level? The substrate?

- In the spirit of making this industrially-manufactured superconducting qubits competitive with existing technologies, can the authors sketch the potential improvement that they can do to push the T1 performance beyond this 300us level? And what about other types of qubit, like fluxonium, is it possible to make such qubits as well in this facility?

(5) The reported qubit frequency (or the JJ resistance) RSD is very interesting and I'm surprised with the still rather-large variation (8%-20% for Rn) across the wafers. Even at the single-die level, the achieved performance (4%-8% for Rn) does not seem to be substantially better than those achieved using the angled-shadow evaporation and metal liftoff technique. The authors mention that it would be possible, in principle, to achieve much better performance using advanced control techniques. Based on the authors best knowledge of this process, how much better would they be able to push down this variation?

(6) Finally, there is this work from the Osborn group in Maryland, looking at the fabrication of transmon qubits using optical lithography on a 300 mm process (<https://iopscience.iop.org/article/10.1088/2058-9565/ab0ca8>). Can the authors comment on how this manuscript is different from the Maryland work?

Author Rebuttals to Initial Comments

We are thankful for the extensive and detailed feedback received by all three referees, allowing us to improve our work significantly. In this document we reply to all questions raised by each referee and mention the corresponding changes made to the manuscript. Our responses are written in blue.

We have made appropriate changes to the manuscript, while trying to maintain a similar word count and keeping the number of additional references minimal, as requested by the editor. Most of the extra material was therefore added to the supplement. Note that we can likely reduce the number of references in the manuscript further if necessary.

Referee #1 (Remarks to the Author):

The authors of this manuscript describe their work on fabricating superconducting qubits using semiconductor CMOS fabrication technology on 300 mm Si wafers. They report statistics of the qubit relaxation and coherence times across the wafer. They conclude that their approach, based solely on optical lithography and removal of materials by reactive ion etching, promises a better way of upscaling of superconducting quantum processors when compared to widely adapted metal lift-off, multi-angle evaporation (such as Dolan bridge and Manhattan techniques), and electron-beam writing.

This work is a continuation of the authors' work published in npj Quantum Information in 2022. [1] The fabrication techniques, characterization of their qubits, and even the structure of the manuscript follows closely their earlier work. To be fair, the authors carried an extensive study to include characterization of hundreds of qubits as well as thousands of Josephson junctions. However, in my opinion, this work **does not** rise to the level of a Nature publication. My conclusion is based on the followings:

We are grateful for the referee's detailed comments on our work and appreciate the acknowledgement of the extensive study that we want to share with the community. We regret that the referee does not find our results sufficiently convincing for publication in Nature. Below we address all raised concerns and hope that our revisions further demonstrate the validity and high impact of our work and can help convince the referee despite the initial reservations.

To be more precise: we understand that there might also be misunderstanding, in that we realize that our work does not yet demonstrate a better way of upscaling – it is rather a first, extensive and (in our view, and, as it appears, also that of the other referees) promising demonstration of an alternative way of upscaling that can leverage several of the advances of the CMOS industry in a reasonable way. We realize that many further optimizations are possible, and only after those have been performed, the community will be able to evaluate which method is the more promising, in the long run. We explicitly adjusted our manuscript to ensure that this message is nuanced, and we hope with the answers to the referee's question to be able to convince the referee of the validity of our alternative approach.

In the manuscript we have rephrased the following sentences to present our work as a promising alternative fabrication method:

Abstract:

This result marks the advent of an alternative and novel, large-scale, truly CMOS-compatible fabrication method for superconducting quantum computing processors.

Third paragraph:

With the scaling requirements of future quantum processors in mind, industry scale fabrication of high coherence qubits utilizing all-optical lithography and exclusively reactive ion etch on 300 mm diameter wafers forms an attractive alternative, since it could fully leverage the state-of-the-art in advanced CMOS fabrication, in line with recent developments in silicon quantum dot qubit fabrication²⁶.

Fourth paragraph in the manuscript, we removed the claim.:

In this work we demonstrate for the first time superconducting transmon qubits fabricated on 300 mm silicon wafers at the foundry-standard cleanroom of Imec, using industry-standard methods that leverage earlier learnings³² and demonstrate the real advantages of advanced CMOS fabrication methods in terms of processing splits, process parameter control and advanced metrology to identify critical process steps.

I.1 - Scaling up the number of qubits is certainly important for quantum information processing. However, maintaining high qubit coherence and relaxation times while scaling up is just as important. The recent development in the field is pushing average T1 times beyond ~ 300 us, several times better than what is achieved in this work. [2-4]

We fully agree with this statement, quantum information processing requires high coherence qubits in large numbers. In this work we demonstrate exactly that it is possible to start with fabrication at scale and optimize towards high coherence. We acknowledge that our qubits do not (yet) surpass the 300 us shadow evaporated hetero devices optimized for decades by the field. As detailed in the works referenced by the referee, these Ta and Nb fabrication processes are highly optimized. Special care went into almost epitaxial sapphire/Ta interface growth, or capping layers to engineer the metal-air interface, techniques, and optimizations we have yet to explore and that are fully compatible with our approach. However, we already come very close with many devices surpassing 100 us, while tremendously exceeding any preceding fabrication volume. We do find this result an extremely encouraging baseline on which our and other facilities will keep on improving, hopefully for decades to come, like the community did for shadow evaporated junction qubits.

To make it more clear in the text that this result is the baseline and more extensive interface engineering optimizations are possible, we have made the following adjustments.

Section “Coherence limitations and interface defects.”, second paragraph:

Projecting the total interface participation to zero yields $\delta_0 = (1.67 \pm 1.69) \times 10^{-7}$, comparable with previously reported values of similar Si substrates^{47,48}. This corresponds to a T_1 limit of $\sim 0.3_{0.15}^{\infty}$ ms at $f_{qb} = 3$ GHz, considerably greater than the mean values depicted in Figure 2a, affirming that, at present, qubit relaxation is predominantly dictated by capacitor interface losses. Interface engineering techniques with metal seed layers, or metal encapsulation⁴⁹ hold promise to further improve on this process.

Section “Conclusion and outlook”, final paragraph:

Looking ahead, the qubit fabrication process presented in this work, aligned with CMOS foundry standards, establishes a robust baseline for future enhancements. The integration of industry-standard techniques - such as chemical mechanical polishing, silicon trenching, and the deposition of sophisticated material stacks - alongside three-dimensional integration, will provide the necessary yield, uniformity, and connectivity for the fabrication of increasingly complex quantum processors.

I.2 - The relaxation times and Hahn-echo times reported in this work are for isolated qubits. It would be interesting to see how these times are holding when these qubits are coupled to each other. Are the authors planning experiments on coupled qubits?

Yes, experiments are being planned on connected qubits, however, to establish required grounding, electromagnetic shielding and signal delivery with negligible crosstalk, we are developing industrial 3D integration methods compatible with superconducting qubit technology. We consider these additional fabrication modules beyond the scope of this work. This point is emphasized both in the introduction and in the new outlook presented in the reply to the question above.

I.3 - There is significant variation of performance between qubits in the center and near the edge of the wafer. No clear explanation of this discrepancy is offered. Is it the variation in dry etching, or Ar milling, or sputtering? Could the authors design experiments to point which processing steps is the likely cause?

We report for the first time on this centre-to-edge dependency in the qubit performance, an observation which, in itself, is enabled by the relative consistency of our approach. At present we do not have the data to confidently appoint the culprit process with certainty, since more and different (and extensive) DOEs would need to be performed to that goal. We will pursue such DOEs in the future, but given the extent required, see them as going beyond the scope of this initial demonstration. Possible sources of loss that are considered to be linked with observed centre-to-edge dependence are patterning (litho + etch) and related surface etch residues, based on SEM inspections (Supplementary Fig. 5ab). Other likely candidate processes are surface cleaning steps, with corresponding centre-to-edge etch residue variation, the argon milling and oxidation process, or the Si wafer bulk losses. Based on blanket tri-layers measured with CIPT (current in-plane tunnelling), we don't see a centre-to-edge non-uniformity in the argon milling + oxidation process itself, however on patterned devices this story might be more complicated and therefore different. An experiment that we will do in a future work is monitoring superconducting resonator Q-factors after every processing step across the wafer to identify when this centre-to-edge loss non-uniformity arises. We do want to stress that such an experiment is worthy of its own manuscript considering the extensive number of measurements that would go into these multiple resonator Q-factor wafer maps.

We now mention the future study with resonator Q-factor measurements in the Supplementary section X, second paragraph:

The identification of the centre-to-edge dependent losses will be the subject of a future study with superconducting high Q-factor resonators.

I.4 - In my opinion, Eq. 1 is simplified without a good justification. Why did they have to use an effective total interface loss δ_t ? How do the individual losses compare to each other at the SA (silicon oxide), SM (some kind of silicide and/or Si oxide), and MA (aluminum oxide)? δ_0 are primarily due to Si substrate bulk losses. How does it compare to published literature?

The reason for this choice is to prevent “over-modelling” interface participation analysis often seen in literature. In the individual interface loss model many assumptions regarding the dielectric constants and thicknesses of the interface layers must be made, directly impacting any claims made about the different interface contributions. Meaningful

disentanglement of the different interface losses can be done by independently varying these interfaces with for example different isotropic and anisotropic etch techniques resulting in different substrate recess and undercuts as done by Woods et al. [1]. However, this is not something we can replicate on different devices on the same wafer. We are very much interested to learn the location of our losses, but only in a meaningful way. With our approach we can at least with confidence separate the combined interface losses from any residual losses without risking “over-modelling”, as we have 4 datapoints, one variable and two fitting parameters δ_t and δ_0 .

The capacitor designs on sub-die DI are made specifically to have a constant cross-section resembling coplanar waveguide geometry. In this case, we can conveniently utilize the analytical formulas for the interface participation ratios derived via conformal mapping theory as shown by Murray et al [2], circumventing the need for more expensive numerical simulations. They show that in the CPW geometry, the participation ratios of the MA, SM and SA interfaces are all proportional to each other, allowing us to group all interface participation ratios into one variable that is a function of the designed widths/gaps of our devices, and an effective total interface loss δ_t independent of the geometry.

Regarding the residual loss $\delta_0 = (1.67 \pm 1.69) \times 10^{-7}$, we appreciate the suggestion to compare this value with literature reports. We find that our reported value does not violate the lower bound set by bulk Si loss reports that we could find in literature $\delta_{bulk} = (1.3 \pm 0.3) \times 10^{-8}$ [3]. We do note that this reported value is for float zone high resistivity Si, while our substrate is grown with the Czochralski method. In other works, with superconducting resonators, Si (> 3500 Ohm cm, Siltronic AG) bulk losses are estimated from participation ratio scaling experiments [1,4] in the range $[1.3; 2.8] \times 10^{-7}$ comparable to our residual loss limit. These comparisons have improved our confidence in the value we report as a lower bound on the loss not located at device interfaces (including the JJ losses, bulk losses, etc).

[1] Woods, W. et al. Determining Interface Dielectric Losses in Superconducting Coplanar-Waveguide Resonators. *Physical Review Applied* 12, (2019).

[2] Murray, C. E., Gambetta, J. M., McClure, D. T. & Steffen, M. Analytical determination of participation in superconducting coplanar architectures. *IEEE Trans. Microwave Theory Techn.* 66, 3724–3733 (2018).

[3] Read A, Chapman B, Frunzio L, Schoelkopf R. The dielectric dipper: experimental comparisons of common dielectric substrates in isolation. *InAPS March Meeting Abstracts 2023* (Vol. 2023, pp. S73-009).

[4] Melville, A. et al. Comparison of dielectric loss in titanium nitride and aluminum superconducting resonators. *Applied Physics Letters* 117, 124004 (2020).

We have included this interesting comparison with literature values in the manuscript. Section “Coherence limitations and interface defects.”, second paragraph:

Projecting the total interface participation to zero yields $\delta_0 = (1.67 \pm 1.69) \times 10^{-7}$, comparable with previously reported values of similar Si substrates^{47,48}, although here also including intrinsic JJ losses. This corresponds to a T_1 limit of $\sim 0.3_{0.15}^{\infty}$ ms at $f_{qb} = 3$ GHz, considerably greater than the mean values depicted in Figure 2a, affirming that, at present, qubit relaxation is predominantly dictated by capacitor interface losses.

I.5 - The authors state in the manuscript “The qubit frequency as function of applied magnetic flux did not exhibit any observable avoided crossings with strongly coupled (> 250 kHz) TLS (data not shown)”. Why not show the data in the supplementary section?

We initially omitted to include this figure because nothing interesting could be seen in the figure, but we agree with the referee that adding this data improves transparency and completeness of our work. We have included Supplementary section IX and Supplementary Figure 9 to the manuscript and updated the main text reference accordingly.

Section “Coherence limitations and interface defects”, third paragraph:

The qubit frequency as function of applied magnetic flux did not exhibit any observable avoided crossings with strongly coupled (> 250 kHz) TLS (Supplementary Fig 9), reminiscent of TLS located inside the JJ barrier^{44,45}.

I.6 - The authors state in the manuscript:

“Our analysis indicates that significant variability is currently coming from the controlled oxidation process, which can still be further fine-tuned. Residual area variance can be addressed with more advanced lithographic control tricks, and etch fine tuning, in principle beyond the range currently achievable with angled shadow evaporation techniques^{3,51}. Furthermore, higher order contributions to the JJ resistance variation e.g. aluminum grain-size effects, and sidewall-edge-slope (impacted by etch and argon milling) could be further addressed. Finally, at design level, larger junction areas (with higher RA values) can be targeted, minimizing junction size variation, consistent with recent literature reports³.”

The authors are discussing potential engineering controls to limit variation across the wafer. Similar engineering controls can be implemented with angled shadow evaporation techniques. I don't see how their technique is better in this respect.

We acknowledge and appreciate the referee's critical stance on our results; however, we would like to stress that this work should not be seen as the end of the line, but rather the gateway to potential continued improvements along a road which is now parallel and synergistic with the transistor technology node highway. Today's fabrication processes leading to the latest transistor technology nodes achieve unprecedented control over entire 300mm wafers (CFET – advanced, gate-all-around transistors linked to the Angstrom level CMOS nodes predicted for the 2030's - target gates with CD's of 14 ± 2 nm and 42 nm pitch [5]), and advanced optimization techniques have also been demonstrated in 'older' technology nodes (e.g. the 65 nm node, first production in 2006/2007) already achieving $<1\%$ CD variation on 140 nm metal patterns across 300 mm wafers [6]. While in this work we are not yet using such advanced processing techniques and optimization cycles (since the primary aim was to demonstrate yielding superconducting qubits with high coherence times), we certainly are able to do so in the future. Shadow evaporation techniques would likely not be able to achieve such scalable fabrication target parameters, as far as we can currently extrapolate. This is also suggested by the non-existence of any angled shadow evaporation tools for 300 mm diameter wafers, which is the industry standard with the most advanced tooling sets and best process control. Grain size and surface roughness can indeed be further improved with deposition condition optimization, deposition temperature, addition of seed layers, addition of impurities and other industrial processes such as planarization with chemical mechanical polishing. The degree of control and methods available in a foundry environment can therefore be significantly larger than in a laboratory employing standard shadow evaporated processing.

While we agree with the implicit message that further work is certainly required, similar limitations of angled shadow evaporation techniques are also noted by the community trying to optimize this approach. In that sense, our work forms, in our view, an interesting and promising alternative with many potential benefits, a feeling that appears to be shared by many in the community. To make the above reasoning a bit more explicit, a recent work was published on the optimization of the angled evaporation of JJs for qubits [7] pointing out

the challenges of this fabrication technique, affirming the value of alternative, subtractive-etch based methods as the one we demonstrate here at scale.

[5] Vincent, B., Boemmels, J., Ryckaert, J. & Ervin, J. A Benchmark Study of Complementary-Field Effect Transistor (CFET) Process Integration Options Done by Virtual Fabrication. *IEEE Journal of the Electron Devices Society* 8, 668–673 (2020).

[6] Qiaolin, Z., Poolla, K. & Spanos, C. Across Wafer Critical Dimension Uniformity Enhancement Through Lithography and Etch Process Sequence: Concept, Approach, Modeling, and Experiment. *Semiconductor Manufacturing, IEEE Transactions on* 20, 488–505 (2007).

[7] Moskalev, D. O. et al. Optimization of shadow evaporation and oxidation for reproducible quantum Josephson junction circuits. *Sci Rep* 13, 4174 (2023).

We have emphasized more strongly in the manuscript that our work presents an alternative approach that can be further optimized.

Third paragraph:

With the scaling requirements of future quantum processors in mind, industry scale fabrication of high coherence qubits utilizing all-optical lithography and exclusively reactive ion etch on 300 mm diameter wafers forms an attractive alternative, since it could, fully leverage the state-of-the-art in advanced CMOS fabrication, in line with recent developments in silicon quantum dot qubit fabrication²⁶

Section “Conclusion and outlook.”, final paragraph:

Looking ahead, the qubit fabrication process presented in this work, aligned with CMOS foundry standards, establishes a robust baseline for future enhancements. The integration of industry-standard techniques - such as chemical mechanical polishing, silicon trenching, and the deposition of sophisticated material stacks - alongside three-dimensional integration, will provide the necessary yield, uniformity, and connectivity for the fabrication of increasingly complex quantum processors.

1.7 - The authors show significant variation in JJ resistance from batch-to-batch and wafer-to-wafer within the same batch, especially for the smallest JJs. Do the authors understand why they observe such a big variation? and how they plan to address this?

We are currently investigating further process improvements (deposition, litho, etch, Ar milling and oxidation) to understand where the variability is coming from. Our current understanding lets us believe that the barrier oxidation step is likely responsible for most of the variability. We plan to extend the barrier oxidation duration to explore the saturation condition of the barrier oxide thickness to a more uniform value across the wafer [8]. The tunnel barrier resistance scales exponentially with the thickness of the oxide, likely explaining the rather large variability between batches and wafers (and likely also on the same wafer). Additionally, we are exploring the impact of tool conditioning.

[8] Jeurgens, L. P. H., Sloof, W. G., Tichelaar, F. D. & Mittemeijer, E. J. Growth kinetics and mechanisms of aluminum-oxide films formed by thermal oxidation of aluminum. *Journal of Applied Physics* 92, 1649–1656 (2002).

We included some of this discussion in the supplementary section VII of the manuscript.

The observed variation in JJ resistances in the wafer-to-wafer and batch-to-batch comparisons could be due to variations in deposition, lithography, etch, argon milling, and barrier oxidation. The barrier oxidation uniformity and reproducibility is likely playing a dominant role, due to the exponential dependence of the tunnel barrier resistance on oxide thickness. A prolonged barrier oxidation to guarantee a more uniform saturated oxide thickness could help. We note that currently our efforts prioritize process optimization in terms of qubit metrics,

while more extensive wafer-to-wafer and batch-to-batch reproducibility studies, including cryogenic qubit measurements, are planned for future works.

1.8 - The authors state in the manuscript:

“Shadow evaporated JJ often show notable parameter aging (increase in the normal resistance as a function of time) with reported values as high as tens of percents, over periods spanning days to weeks^{3,52}. Overlap junctions fabricated in this work, however, show significantly less aging, limited to a decrease of 3.8% over 146 days”

Again aging in Shadow evaporated JJs could be addressed in the fabrication by introducing a gentle O₂ descumming step. [5]

While there is a process that can stabilize shadow evaporated JJs, it is not clear from published work whether such a process stabilizes aging while maintaining high coherence times of superconducting qubits. We added another citation to a recent (2022) published work from the Siddiqi group at UC Berkely [9] that clearly shows established groups still notice significant JJ aging in their devices, suggesting that the O₂ descumming solution is for some reason not adopted as a solution to the aging problem for shadow-evaporated superconducting qubits.

[9] Kim, H. *et al.* Effects of laser-annealing on fixed-frequency superconducting qubits. *Appl. Phys. Lett.* **121**, 142601 (2022).

1.9 - The authors focus on qubits based on Al. As discussed before, there are other material platforms promising better qubit coherence times. Could they incorporate these emerging materials into their processing? If so, how? There is also recent work on Nb trilayer junction qubits also based on all optical lithography. [6] With much larger superconducting gap compared to Al, Nb qubits could be less prone to quasiparticle loss. Could the authors comment on how Nb trilayer JJ qubits compares to their qubits?

[1] npj Quantum Information (2022) 8:93

[2] Nature Communications volume 12, Article number: 1779 (2021)

[3] npj Quantum Information volume 8, Article number: 3 (2022)

[4] arXiv:2304.13257

[5] J. Vac. Sci. Technol. B 30, 010607 (2012)

[6] arXiv:2306.05883

Yes, one of our next development cycles will include combining different materials with larger superconducting gap. The initial process is planned to be very similar to how it is typically done with shadow evaporation approach where Al/AlO_x/Al JJ is fabricated on top of the larger superconducting gap material circuitry. We have been extensively exploring this route for some time where we identified potential pitfalls and solutions when combining other materials such as Nb with the processing required for overlap junction fabrication [10].

The paper the referee mentioned is a very interesting work that is appreciated by a lot of groups who were trying in the past to use Nb trilayer JJ for high-coherence superconducting qubits, albeit with little success due to challenging sacrificial oxide removal selective to the barrier oxide (including work done by ourselves, we refer to [11] for more details). While the Chicago/Stanford group succeeded and demonstrated high coherence times, they also

mention that the process window for such an approach is very narrow which significantly affects the yield, variability and reproducibility of the process. For this reason, and our own previous efforts related to this, we at present do not believe that such process will be the most promising one for large scale integration of superconducting qubits.

[10] Van Damme, J. et al. Argon-Milling-Induced Decoherence Mechanisms in Superconducting Quantum Circuits. *Phys. Rev. Appl.* 20, 014034 (2023).

[11] Wan, D. et al. Fabrication and room temperature characterization of trilayer junctions for the development of superconducting qubits on 300 mm wafers. *Jpn. J. Appl. Phys.* 60, SBBI04 (2021).

In the manuscript we have made the following related changes:

Third paragraph:

While CMOS foundry-compatible²⁸⁻³⁰ processes and qubits fabricated partially in a foundry environment³¹ have been shown before, no 300 mm wafer-scale qubit fabrication has thus far been presented.

Referee #2 (Remarks to the Author):

Report on High-coherence superconducting qubits made using industry-standard, advanced semiconductor manufacturing, Van Damme et al.

Van Damme et al. demonstrate the fabrication of superconducting aluminum transmons on 300-mm wafers using essentially a CMOS line and toolset. The resulting Josephson junctions are tested at room temperature and show 5-8% cross-wafer variation, with high-90's hard yield (works versus doesn't work). The qubits exhibit coherence times in the 100-200 us range for the center of the wafer, reducing to 40 us towards the edge in a usual "donut" shape. The authors characterize and observe effects seen in the qubit community, including fluctuating TLS loss at the capacitor.

The notable accomplishments are:

1) Demonstrating reasonable transmon qubits (coherence and yield) in a 300-mm line using 193-nm optical immersion lithography, instead of the usual E-beam lithography.

2) Use of a "trilayer-type" Josephson junction formed with subtractive etching in multiple steps, instead of the usual single-step additive Dolan-style or Manhattan-style junction with lift-off.

3) Since the additive JJ approach does not have an exposed edge, the observed aging effect is much less than is typically seen with Dolan or Manhattan JJs, which generally have exposed edges.

4) Process monitoring of the junctions and qubits across a wafer with statistical characterization both at room temperature and a cryogenic temperatures.

I think this is wonderful to see, as having a commercially accessible foundry be able to produce high-quality and relatively high-yield superconducting devices is going to be a big advantage to academic groups (if they can afford it) and companies worldwide. As the authors say, the next steps would be to demonstrate 3D integration and heterogeneous packaging technologies that are also relatively mature in such foundries.

I very much appreciate the hard work that went into demonstrating the above results and congratulate the team for doing it. It is impressive to see several complex steps come together.

We are very happy to read that the referee shares our enthusiasm in the achieved results and the exciting outlook towards 3D integration at scale. We appreciate greatly the time and effort that went into the thorough and comprehensive summary of the main accomplishments of our work. Below we comment further on the “blue team” and “red team” remarks.

Whether this is worthy of Nature is an editorial decision. In the spirit of reviewing the manuscript, here are a few “blue-team” and “red team” comments for the editors and authors to consider:

On the “blue team” side, in terms of novelty, I would say that the main innovation is the dome-shaped “trilayer” junction, which features:

2.1 - fabrication using multiple steps of subtractive etching. This can be contrasted with the usual Dolan-bridge or Manhattan-style approaches, which are generally additive approaches that use lift-off processing and generally a single deposition and oxidation step. Lift-off is known to be less reproducible and scalable than a subtractive etch approach.

2.2 - High coherence times, despite fabricating the junction in several steps. The bottom electrode is patterned and etched. Then, the metal surface is cleaned in an Ar plasma, and the clean surface oxidized. And then the top electrode is deposited and etched.

We agree with the referee that the high coherence times of our overlap style JJ qubits fabricated at (yielding) scale is the main merit of our work. Our primary goal was to push relaxation and coherence times on our devices and unlock CMOS lines for future quantum information processor fabrication. And obviously, further optimization is possible and needed, we agree with the other referees that this is only the first step in this direction.

2.3 - The “dome” shape isolates and protects the junction oxide from further oxidation (to some degree). This is likely one reason these devices age much less than conventional superconducting qubits. Note that previous Josephson devices made from niobium trilayers, both qubits and traveling wave parametric amplifiers, similarly do not age due to an anodization step that surrounds and protects the junction barrier. However, this adds an additional lossy oxide that tends to impact the qubit coherence. That additional anodization step is not needed with a “dome” junction. On the other hand, the lateral sides of the dome are still exposed to some degree, similar to a Manhattan junction. Late-stage static, dynamic, or plasma oxidation is sometimes used to “seal” the edges with AlO_x, and that may also be at play here. Whatever the reasons, the dome junction is an important innovation (but, see also below).

This is an interesting comment, likely in line with the remark of referee #1 who pointed out that JJ aging stabilization can be achieved by O₂ descumming. But we agree with the referee that additional plasma oxidation could be adding unwanted lossy oxide, likely compromising qubit coherence. Our approach does not require any such compromise and seems natively robust against aging. While it was initially, also on our end, an unexpected result, we share the sentiment that this is a clear advantage of this route.

Mixed “blue-red”:

2.4 - However, the “dome” junction adds morphology to the fabrication. It requires precise control of a rounded dome that supports the junction barrier, for example, as well as control of the etching in to the silicon substrate. Congratulations to the team for getting it to work at close to the 99% yield level. However, we know that such morphology can be problematic when trying to achieve yields with more 9’s. This is one reason for the advent of planarization and planarized processing in the CMOS industry, particularly as the number of layers and complexity increases. This is something that the IMEC team will need to address as they go forward. Presumably, the motivation for 300-mm fab is to fabricate chips that contain thousands to millions of qubits, and so the yield needs to be at the six 9’s level (or more). And, at those numbers of qubits, more wiring layers and integration will be required. It will be interesting to see if the dome is compatible with such high yields and complexity.

The referee makes a great point here, we also believe that the dome morphology of the junction introduces additional etch and argon milling uniformity challenges. We are actively working on optimizations towards even higher yield and uniformity of the JJ area by advancing the dome and exploring other promising morphologies, including indeed potential planarized ones that aim to leverage CMP and explore its limitations for superconducting qubits. 3D integration, in particular In-bump bond flip-chip stacking, would in principle allow scalable integration of out-of-plane non-planarized JJ morphology in the qubit layer, while provide the necessary multilayer wiring on the opposing chip. Nevertheless, this is outside the scope of this manuscript and will be discussed in detail in upcoming studies.

On the “red team” side (and with the caveat that this work is likely only the first step of many to be taken):

We acknowledge that this result is a solid basis for an alternative superconducting quantum circuit fabrication approach that has many more advancements to come and we thank the referee for pointing this out. We have strengthened the emphasis of this work as a baseline for further improvements in the manuscript.

Section “Coherence limitations and interface defects.”, second paragraph:

Interface engineering techniques with metal seed layers, or metal encapsulation⁴⁵ hold promise to further improve on this process.

Section “Conclusion and outlook”, final paragraph:

Looking ahead, the qubit fabrication process presented in this work, aligned with CMOS foundry standards, establishes a robust baseline for future enhancements. The integration of industry-standard techniques - such as chemical mechanical polishing, silicon trenching, and the deposition of sophisticated material stacks - alongside three-dimensional integration, will provide the necessary yield, uniformity, and connectivity for the fabrication of increasingly complex quantum processors.

2.5 - Most, if not all, of the processing steps have been demonstrated before in places like IBM, Lincoln Laboratory, and elsewhere, primarily on 200-mm wafers, although generally not published in this level of detail for various reasons (IP, trade secrets, etc.). I do not see a big difference between 200-mm and 300-mm wafers. The toolset is basically the same, and the advances made in the newer 300-mm tool technology are generally available in 200-mm bridge tools.

The fabrication capabilities of Lincoln Laboratory and IBM are state-of-the-art and set the benchmark. Their approach is an “industrialization” of the laboratory developed techniques still relying on angled shadow evaporation and liftoff. As pointed out by the referee, these techniques do exist on 200 mm tools, but have been abandoned for the high technology nodes (the vast majority of which, if not the entirety, is executed on 300 mm processes), due to the superior dimension control and yield achieved with subtractive etch methods. We are not saying that pushing 200 mm bridge tools and potentially developing new tools specifically for advanced quantum information processor fabrication is a bad idea, on the contrary, our approach now offers an alternative angle of approach directly leveraging the decades of investments and dedicated optimization that already went into the more advanced 300 mm tooling sets, while still achieving high coherence qubits, against the expectations of many in the field believing that JJ barrier argon milling and subtractive etch damage is detrimental. A recent work was published on the optimization of the angled evaporation of JJs for qubits [1] pointing out the challenges of this fabrication technique, affirming directly our statements.

We also refer to our reply to comment 1.6 of referee #1.

[1] Moskalev, D. O. et al. Optimization of shadow evaporation and oxidation for reproducible quantum Josephson junction circuits. *Sci Rep* 13, 4174 (2023).

2.6 - I have seen public talks of high-coherence qubits using 193-nm lithography on 300-mm wafers from Albany NanoTech pre-COVID, and I am aware of unpublished works at other facilities. While 193-nm lithography is certainly faster (more manufacturable) than E-beam lithography, it is also much more expensive to implement (mask cost primarily) and slower to iterate (re-spinning masks). Nonetheless, I am sure that IMEC could similarly use E-beam lithography as well in this process.

We are aware of the pre-COVID published work from SUNY Polytechnic in Albany [2], which is interesting work demonstrating the scaling advantage of optical lithography over e-beam writing. However, we would like to point out that in their work Josephson junctions were not fabricated on the wafer scale. After the lithography of the base metal layer and JJ resist patterning, the 300 mm wafer was diced into coupons, of which only a few were then processed, using the conventional angled shadow evaporation and lift-off techniques, incompatible with a CMOS line. Additionally, only two functional qubits are reported with moderate relaxation times of $T_1 < 30$ μ s (on a 3D transmon design that has smaller interface participation ratios than even our largest devices). This is in stark contrast with the large-scale statistics and performance we achieve across devices fabricated on the entire wafer.

We acknowledge the flexibility advantage of e-beam lithography in terms of design iterations, and we certainly utilize e-beam lithography ourselves for initial explorations. However, once a set of mature device designs is available, process splits and optimizations benefit greatly from the speed-up and stability offered by optical lithography, quickly recovering the initial mask cost.

We added a sentence at the end of the second paragraph in the introduction:

While CMOS foundry-compatible²⁸⁻³⁰ processes and qubits fabricated partially in a foundry environment³¹ have been shown before, no 300 mm wafer-scale qubit fabrication has been presented.

[2] Foroozani, N. et al. Development of transmon qubits solely from optical lithography on 300 mm wafers. *Quantum Sci. Technol.* 4, 025012 (2019).

2.7 - I am not aware of anyone in the above point using 193-nm *immersion* lithography. The immersion is used to obtain finer feature sizes, typically at the CMOS 65-nm node and below. I'm glad to see that the team has done this. However, despite the added fab complexity, it apparently did not improve the junction uniformity over other published results. This may not be too surprising, since much of the critical-current non-uniformity is speculated to arise from non-uniformity in the junction barrier.

Indeed, the feature size capabilities of the 193-nm immersion lithography support more advanced CMOS nodes than the non-immersion 193-nm lithography. We would like to point out that, at present, we do not yet leverage all the capabilities of this technology, for example, optical proximity corrections, leaving plenty of room for improvements down the line. In addition, our analysis indeed points towards significant contributions from critical-current non-uniformity that should be addressed independently of the patterning optimization, for example by prolonged barrier oxidation (see also our reply to referee #1 question 1.7). Playing red-team ourselves, some of the lithography precision is also compromised by the edge slopes due to etching followed by the argon milling rounding to achieve the dome-shape mentioned by the referee. This is essentially a trade-off between yield (improved step coverage) and area uniformity (edge slope variation). However, we are confident that optimization towards a sweet spot is promising, and we have additional ideas to overcome this trade-off altogether.

2.8 - Almost all standard materials used in superconducting qubits are CMOS compatible, with perhaps the sole exception being gold pads that some groups use, which is easily replaced with other high-conductance metals like platinum. I don't see this as a big differentiator. The main differences encountered in a CMOS foundry is that the temperatures generally used in CMOS fabrication are higher than would generally be allowed for superconducting qubit (JJ) fabrication. It does take process development to figure these steps out, e.g., how to bake the resist uniformly at lower temperatures, but that is technical process development.

The referee points out a valid challenge in the reduced thermal budget for qubit fabrication. However, we share the sentiment that this is a question of proper process development within the thermal budget. Our work demonstrates that this is possible.

Referee #3 (Remarks to the Author):

The manuscript reports the fabrication of superconducting qubits (transmons) with relatively high coherence (best T1s are more than 100us) in their own 300 mm advanced CMOS manufacturing facility. The reported statistics on yield, variability, and aging behaviour is very informative, particularly for researchers with limited access to advanced facilities like this and (understandably) little control over the manufacturing steps. The technical achievement reported in this manuscript is also very timely, the superconducting hardware community is currently pushing for devices beyond >100 qubits, and being able to harness

some of the advantages of large-scale manufacturing while achieving high coherence will certainly be useful. There are numerous challenges that have to be overcome and the authors have done an impressive work surpassing them. As far as I know from published literatures, the reported T1 level is the first from such a facility, and hence will set the bar to surpass for the rest of advanced 300 mm manufacturing facilities.

The authors seem to have analysed their results in great detail, and present some of these analyses in the main manuscript. However I don't think that the presentation has been done in a way that is clear to the reader, and the majority of my feedback is related to this. I can recommend a publication in Nature after the issues/questions below have been addressed in a satisfactory manner (for me, and the readers in general).

We express our sincere appreciation to the referee for their comprehensive and insightful evaluation of our study. Their acknowledgement of our findings and contributions is deeply valued. We have done our best to address all the comments below and we believe that, with the implemented changes, we have significantly improved the completeness and quality of the manuscript.

(3.1) The authors claim to “*demonstrate the real advantages of advanced CMOS fabrication methods in terms of processing splits, process parameter control and advanced metrology to identify critical process steps*“. I have difficulty convincing myself that this is already described in the manuscript.

- Which parts of the manuscript show that they have made use of the process parameter control and advanced metrology to identify critical process steps?

We thank the referee for pointing out this hyperbole in the manuscript. Significant process development went into this first demonstration with a number of process splits that were inspected both in-line and out of the fab environment (e.g. using SEM, CD-SEM, optical, TEM, junction resistance, resist stacks optimization, over-etch monitoring, etc.). Nevertheless, we do not yet leverage or demonstrate the full extent of process control and reproducibility capabilities of these CMOS line tools. We agree that this work should be shared as a baseline for foundry like qubit fabrication and is not yet the end of the line. We have adjusted the manuscript accordingly.

Fourth paragraph in the manuscript, we removed the claim.:

In this work we demonstrate for the first time superconducting transmon qubits fabricated on 300 mm silicon wafers at the foundry-standard cleanroom of Imec, using industry-standard methods that leverage earlier learnings³² and demonstrate the real advantages of advanced CMOS fabrication methods in terms of processing splits, process parameter control and advanced metrology to identify critical process steps.

- Furthermore, it looks like it is precisely the opposite that happened: the authors seems to have observed radial dependence in the qubit best T1s (figure 2a) as well as the frequency variation (figure 4c). I would have hoped that industrial manufacturing facility has better control over their processes, which in principle means that variations like this should not have happened in the first place?

This is the very first demonstration of high coherence time qubits fabricated with a CMOS pilot line on a 300 mm wafer. The development just started; we are confident that centre-to-edge variability can be addressed and improved as we know from CMOS technology. We would also like to point out that those centre-to-edge effects in relaxation times are observed and reported for the very first time here and are a natural consequence of the large wafer area and high-level of processing control over the entire 300 mm wafer on which superconducting qubits have never been fabricated before. The fact that we observe this dependence gives us also a unique opportunity to study processing effects on qubit coherence times across a single wafer with a good control over already small fabrication parameter drifts. Our focus in this exploratory work was to demonstrate that industry-standard facilities are capable of fabricating large volume and yield of high-coherence superconducting qubits. Device parameter targeting, variability, including centre-to-edge variation, although already respectable, are being addressed at our facility in the next stage of development.

(3.2) The authors choose to quote the “time averaged energy relaxation times ($\langle T_1 \rangle_t$) of best performing qubits from sub-dies D1”. I have difficulty in understanding this choice.

- Why are only best performing qubits relevant in this discussion as opposed to the overall T1 performance of all measured qubits? In the context of this manuscript, given the number of dies that can be produced, it is reasonable to expect that a substantial part of the measured T1 values are also reasonably good and that is perhaps what the authors should also describe to the reader.

We thank the referee for pointing out that the motivation for the representation of our data in this way was not made sufficiently clear in the manuscript.

First, we want to emphasize that only a subset of the 400 measured qubits was designed for high relaxation and coherence times. Most of the devices have designs with intentionally enlarged interface electric field participations to monitor the impact of these interfaces, while some also show Purcell limited relaxation times (mostly the flux tunable qubit designs).

The main differentiation of our fabrication process with respect to the state-of-the-art in literature is the overlap JJ. By monitoring the best devices on each die we are showcasing the upper bound on T1 across the wafer, reducing as much as possible the contribution of TLS loss at capacitor interfaces in an attempt to benchmark the quality of the junctions in our process. This map of measured maximal T1 values can be interpreted as a lower bound on the T1 of the JJ induced relaxation across the wafer, and an attempt to decouple the (more random) effects of TLS losses.

Additionally, the random frequencies of these TLS can result in unlucky defects resonant with the qubits, causing a dominant non-systematic loss channel, known by the community and not the target of this work. It is only when minimizing this random contribution of unlucky resonant TLS as much as possible (by taking the best performing devices) that we can notice the more systematic underlying location dependence on our T1 upper bound. As the referee pointed out, that was not clear in the manuscript.

We have therefore adjusted the manuscript to make our choice for showing the best devices clearer.

Section “Qubit coherence.”, first paragraph:

A total of 394 out of 400 qubits (yield = 98.5%), of a variety of different designs and sizes, were functional and fully characterized (Supplementary Section X for all data). Multiple loss channels contribute to the qubit relaxation times⁴. A lower bound on the intrinsic JJ relaxation time of our process can be set by monitoring the time averaged relaxation times ($\langle T_1 \rangle_t$) of the best performing qubits from sub-dies D1 across the wafer. A centre-to-edge dependence is observed with the highest measured value $\langle T_1 \rangle_t^{\text{max}} = 113 \mu\text{s}$ close the centre, down to the lowest measured value $\langle T_1 \rangle_t^{\text{min}} = 42 \mu\text{s}$ near the edge and with a median of $\langle T_1 \rangle_t^{\text{med}} = 75 \mu\text{s}$ (see wafer map in Figure 2a).

See also discussion in the newly added Supplementary Section X.

- The notation $\langle T_1 \rangle_t^{\text{max}}$ and $\langle T_1 \rangle_t^{\text{min}}$ can be confusing. First of all, the specifiers “max” “min” are only referring to the fact that the $\langle T_1 \rangle_t$ values close to the center of the wafer are the highest and the ones close to the edge are the lowest? Can I instead suggest “center” and “edge”? Am I correct to state that these are measured values and not obtained from the interpolation?

The referee is correct in stating that these are measured values. The max and min values refer to the largest and lowest measured T1 values (time average) of the map in Fig 2a, which happened to be located close to the centre and edge respectively. The rephrasing of the first paragraph of the “Qubit decoherence.” section in the manuscript shown in the previous question should also clarify this comment of the referee.

- The 2D colormap plot, I have to admit, is not very clear. How do typical interpolated curves look when they are plotted in a 1D lineplot?

We thank the referee for pointing out the room for improvement in the visualization of the 2D colormaps. We have updated main text Figures 2a and 4c with a new background colour that is now representing a smooth best fit with a heuristic Gaussian model. We show below the difference between the interpolated data that was used before and the gaussian model fit that is used now. We hope this addresses the referee’s comment.

- Do the authors observe identical or similar center-to-edge dependence when considering the average T1 per die or the lowest T1 per die?

As clarified above, the centre-to-edge dependence is primarily noticeable in our best devices where the convolution with dominant loss from random/unlucky TLS resonant with the qubits is minimal.

We have included this discussion in Supplementary section X and included wafer maps of all measured fixed frequency qubits, separated by their capacitor geometry or surface participation ratio (Supplementary Figures 11 and 12).

- It will be helpful for the reader to see the histogram of T1, T2* (or T2E) of all qubits that are measured in this wafer. The idea is to inform the readers how many qubits (out of 394) are actually crossing the 100us threshold. I appreciate the amount of work that has gone into this manuscript, I understand the challenges, and I am not looking to criticise the work if the majority of the T1 values are not above 100us.

We agree with the referee that an overview of all performed coherence measurements was missing and we have included a new supplementary section X with all this data. Due to the different capacitor geometries of the measured devices (resulting in different interface loss participations, which was the very driver thereof: DOEs to extract junction behaviour), we found cumulative distribution function visualizations more informative (the equivalent histograms were too messy with all the overlapping distributions). We hope the referee agrees with this alternative data representation.

- Finally, what are the frequency range of these qubits? Figure 3a shows the inverse of the qubit quality factor, can the authors states the quality factor as well in the text. This will be helpful as a quick reference for the reader.

Nominal qubit frequencies are printed in Table I and span a frequency range [2.9 GHz ; 6.1 GHz]. Actual qubit frequencies are on average 639 ± 21 MHz higher than designed due to the a priori unknown JJ critical current values of our process (at the time of optical mask design).

We have included the following change to the manuscript to accommodate this request.

Section “Coherence limitations and interface defects.”, first paragraph:

The transmon qubits on sub-dies D1 were designed with four capacitor geometry sizes, resulting in a six-fold difference in the extent of calculated electric-field-energy participation ratios (EPR) at qubit capacitor interfaces (metal-air, substrate-metal, and substrate-air)⁴². This allows for the differentiation between loss sources located at capacitor interfaces from other losses such as bulk substrate loss and more importantly losses intrinsic to the JJ. Qubit energy loss ($1/Q$) scales linearly with the sum of all interface EPR (Figure 3a, with $Q = \{1.7, 1.1, 0.84, 0.42\}$ million). A linear loss model fits the data well.

We have also provided the Q-factor wafer maps of all measured qubits in Supplementary Figures 11 and 12.

(3.3) In figure 3a, the authors plot the inverse of the measured quality factor as a function of the simulated electric-field participation ratio of the interface (let's denote it as $p_{\text{interface}}$), fit it with a linear curve, extrapolate the expected quality factor when $p_{\text{interface}}$ becomes zero, and claim that the inverse of this extrapolated value represents that the T1 limit (around 300 us) that could be achieved if the $p_{\text{interface}}$ is actually zero.

- Am I correct in understanding the reasoning here?

Yes, the referee is correct here.

- Am I also correct in stating that the linear relation is only valid if we assume that the participation ratio in the bulk volume (p_{bulk}) is unaffected when $p_{\text{interface}}$ goes to zero? I'm not entirely sure in which situation this ($p_{\text{interface}}$ goes to zero) would be possible. Even if we assume that it is possible, p_{bulk} should be affected as well and it is not straightforward to state that the linear relation remains valid for zero $p_{\text{interface}}$. Thus, the 300us limit may not mean much. I invite the authors to explain this part.

The referee is mostly correct here, in practice the participation ratio of the bulk would slightly increase in the limit of vanishing interface participation ratio (with some remaining percentages of the fields in vacuum and the JJ). However, this is a negligible relative change because the bulk participation was already 91.8% for our smallest devices (for our largest device, it changed to 92.1%). Additionally, the assumption in the model is that the total residual loss is independent of capacitor geometry, which besides (almost constant) bulk losses also includes the intrinsic JJ losses, spurious junction loss, and possibly other capacitor geometry independent losses. Especially considering the scatter in the data, we are confident our simple linear model is justified.

With this analysis we find that qubit energy relaxation times are limited by the capacitor interfaces losses. This observation is important first because it is comparable with similar findings in the literature for standard shadow-evaporated junctions and second as it directs the future optimization efforts.

To make the motivation of this analysis clearer for the reader, we improved the text in the manuscript.

Section "Coherence limitations and interface defects", first paragraph:

This allows for the differentiation between loss sources located at capacitor interfaces from other losses such as bulk substrate loss and more importantly losses intrinsic to the JJ.

Supplementary section "III. Interface participation ratio", end of first paragraph:

The calculated interface participation ratios for the four different qubit geometries are plotted in Supplementary Figure 3. The corresponding substrate participation ratios are {91.8%, 92.0%, 92.1%, 92.1%}, and the vacuum participation ratios are {7.93%, 7.86%, 7.82%, 7.79%}.

(3.4) Suppose that, for the moment, we agree that the 300us value is a meaningful limit.

- Do the authors have an idea of what is limiting the performance at the 300us level? The substrate?

At this point we cannot separate between the intrinsic JJ loss, spurious JJ loss, and the bulk loss contributions. All can limit the performance. A separation between the bulk and the JJ effects could be revealed by performing the participation ratio study with high-Q resonators that have the same sensitivity to dielectric interface and bulk loss however, they do not contain a JJ. Such study is not straight forward since obtaining comparable participation ratio parameter range would require coplanar-waveguide resonators with large central trace width of 90um which invites magnetic vortex formation despite extensive magnetic shielding or the use of lumped element resonators which has more complex and therefore less accurate participation ratio estimation [2,3,4]. We did perform such analysis on CPW resonators with a smaller PR range with central traces between 2um and 20um, however, results were inconclusive due to relatively large single-photon level intrinsic Q-factor variation. The study with larger range in surface participation ratio will be a part of future work.

[2] Verjauw, J. et al. Investigation of Microwave Loss Induced by Oxide Regrowth in High-Q Niobium Resonators. *Phys. Rev. Applied* 16, 014018 (2021).

[3] Van Damme, J. et al. Argon-Milling-Induced Decoherence Mechanisms in Superconducting Quantum Circuits. *Phys. Rev. Appl.* 20, 014034 (2023).

[4] Crowley, K. D. et al. Disentangling Losses in Tantalum Superconducting Circuits. *Phys. Rev. X* 13, 041005 (2023).

- In the spirit of making this industrially-manufactured superconducting qubits competitive with existing technologies, can the authors sketch the potential improvement that they can do to push the T1 performance beyond this 300us level? And what about other types of qubit, like fluxonium, is it possible to make such qubits as well in this facility?

Based on our participation ratio scaling results the immediate action is to improve capacitor surfaces and interfaces by exploring different types of surface cleans and reduce impurities as well as improve quality of surface and its oxide. Interface engineering with metal liners/seed layers and cappings are also considered (see also reply to referee #1 comment 1.9). Improving losses associated with the substrate can be done by using higher-resistance substrates which host less p-type defects [5]. We are currently evaluating different high-resistivity 300mm wafers in terms of defect density and industrial handling requirements. If after improving these aspects we start being limited by losses associated with the Josephson junction, the first step would be to short the large spurious junction that is always formed in the overlap or shadow-evaporation process. While there are several techniques for spurious junction bandaging in the shadow-evaporation process, we are exploring similar methods for the overlap junction process. After that, there are a number of overlap junction fabrication parameters that can be optimized to minimize JJ intrinsic loss such as the Ar-milling, and oxidation conditions. Additionally, our reactive dry-etch process would allow for deep substrate trenching around the JJ, offering additional electric field participation ratio engineering options.

Other types of superconducting qubit are certainly possible to be implemented. We chose transmon qubits due to its popularity in order to better compare performance metrics with literature. We can however design any other type of qubits with increased complexity. We would like to also stress that the overlap junction process can fabricate both small and large junctions with the same process without the need for parameter or junction geometry optimization based on the junction size.

[5] Zhang, Z.-H. et al. Acceptor-induced bulk dielectric loss in superconducting circuits on silicon. Preprint at <https://doi.org/10.48550/arXiv.2402.17155> (2024).

(3.5) The reported qubit frequency (or the JJ resistance) RSD is very interesting and I'm surprised with the still rather-large variation (8%-20% for R_n) across the wafers. Even at the single-die level, the achieved performance (4%-8% for R_n) does not seem to be substantially better than those achieved using the angled-shadow evaporation and metal liftoff technique. The authors mention that it would be possible, in principle, to achieve much better performance using advanced control techniques. Based on the authors best knowledge of this process, how much better would they be able to push down this variation?

Based on our current analysis, we conclude that the barrier oxidation uniformity is contributing significantly to the reported R_n RSD values, clearly visible by the plateau in the

RSD at large JJ areas in Figure 4a (~8% at wafer level and ~4% at single die level). We know from tri-layer junctions (Nb/Al/AlO_x/Nb) with similarly grown AlO_x barriers at our facility that we can reach Rn RSD < 0.8% for large area junctions [5] (an upper bound for the RA variability). We are confident that with optimization in dimension control (optical proximity effect corrections, resist stacks, etch conditions, post-exposure bake) and bottom electrode morphology control (edge slopes, grain sizes, surface roughness) we can bring the area variability significantly down. With optimized 193nm immersion lithography imec can routinely achieve sub-nanometre standard deviations on the critical dimension of Si etched trenches, more advanced technology nodes (utilizing EUV, high-NA EUV) could potentially push this even lower. We believe an optimized optical lithography and dry-etch approach on aluminum (combined with aluminum grain and roughness optimizations) should reach similar values. We are hopeful to ultimately reach an Rn RSD of reasonable junction sizes below 1.0%, which is within the required range of 0.5-1.0% resistance variability targeted by fixed frequency qubit processor architectures [6] without need for laser annealing.

[5] Wan, D. et al. Fabrication and room temperature characterization of trilayer junctions for the development of superconducting qubits on 300 mm wafers. *Jpn. J. Appl. Phys.* 60, SBBI04 (2021).

[6] Hertzberg, J. B. et al. Laser-annealing Josephson junctions for yielding scaled-up superconducting quantum processors. *npj Quantum Inf* 7, 1–8 (2021).

We have included this discussion in the supplementary section VIII of the manuscript.

(3.6) Finally, there is this work from the Osborn group in Maryland, looking at the fabrication of transmon qubits using optical lithography on a 300 mm process (<https://iopscience.iop.org/article/10.1088/2058-9565/ab0ca8>). Can the authors comment on how this manuscript is different from the Maryland work?

The work from the Osborn group in Maryland is according to us the same work Referee #2 mentioned as the “Albany NanoTech pre-COVID talk” in comment 2.6 which we addressed in detail. In essence the main difference is that, in their work, they immediately dice the 300 mm wafer into coupons and finalize device fabrication on a small scale in the conventional laboratory style method using shadow evaporation and lift-off. They showcase the speedup and volume advantage of optical lithography over e-beam writing, but the similarity with our work stops there. We do the full all-optical processing on 300 mm scale and only dice our finished devices for measurement, while achieving a qubit performance far greater than what is reported in that work.

We added a sentence at the end of the second paragraph in the introduction:

While CMOS foundry-compatible^{28–30} processes and qubits fabricated partially in a foundry environment³¹ have been shown before, no 300 mm wafer-scale qubit fabrication has thus far been presented.

Additional changes

In addition to the changes made upon request of the referees, we have corrected typos and errata as summarized below.

Green: new addition

Orange: Changed value

Red: removed

References

- Bravyi, S. et al. High-threshold and low-overhead fault-tolerant quantum memory. *Nature* **627**, 778–782 (2024).
- Kim, H. et al. Effects of laser-annealing on fixed-frequency superconducting qubits. *Appl. Phys. Lett.* **121**, 142601 (2022).
- Kono, S. et al. Mechanically induced correlated errors on superconducting qubits with relaxation times exceeding 0.4 ms. *Nat. Commun.* **15**, 3950 (2024).
- Neyens, S. et al. Probing single electrons across 300-mm spin qubit wafers. *Nature* **629**, 80–85 (2024).

Typos

- Section “Qubit coherence”, first paragraph:
The time averaged energy relaxation times ($\langle T_1 \rangle_t$) of the best performing qubits from sub-dies D1 show a centre-to-edge dependence with $\langle T_1 \rangle_t^{\max} = 113 \mu\text{s}$ close the centre and down to $\langle T_1 \rangle_t^{\min} = 42 \mu\text{s}$ near the edge with median of $\langle T_1 \rangle_t^{\text{med}} = 75 \mu\text{s}$ across the wafer (see wafer map in Figure 2a).
- Caption of Figure 3:
Inset: the pulse sequence used to scan the spectrum.
- Section “Qubit frequency variability and aging”, second paragraph:
It is therefore prudent to monitor the variability of the JJ area, and the resistance-area product (~~RA = ρd~~) which encapsulates resistivity and thickness into a single parameter, RA, representative of the overall oxidation uniformity (Supplementary Figure 5c).
- Section “Qubit frequency variability and aging”, third paragraph:
Approximately half of the variability can be attributed to ~20% centre-to-edge decrease of the average R_n (see Supplementary Figure 5d).
- Caption of Figure 4:
JJ test structure normal resistances as a function of estimated junction area for a subgroup of 216 devices measured 5 days after fabrication (t_0) and once more 146 days after fabrication. The right y-axis shows the corresponding relative average resistance change for each JJ area.
- Supplementary section “VIII. JJ normal resistance relative standard deviation analysis.”
Modeling this area dependence can be leveraged to disentangle variance contributions of area and barrier uniformity (tracked with the resistance area product parameter ~~RA = ρd~~).
- Supplementary section “VIII. JJ normal resistance relative standard deviation analysis.”
The area of the JJ fabricated in this work includes the rounded surface of the bottom electrode and is described by the formula in the legend of Supplementary Figure 2e.

Errata

Upon further examination by the authors, we identified an error in the originally reported values of the top electrode critical dimensions extracted from CDSEM data (Supplemental Figure 2d). This mistake propagated to the area estimation of the Josephson junctions in main text Figure 4. Below we summarize the corrections made to the original work.

- Section “Qubit frequency variability and aging.” 4th paragraph
This analysis reveals an $RSD_{RA} = 4.864.47\%$ and $\sigma_A = 0.002170.00334 \mu\text{m}^2$ on a single die, meaning that, for all JJ with $A > 0.0450.075 \mu\text{m}^2$, the barrier non-uniformity is the dominant cause of f_{qb} variability.
- Figure 4.
- Section “Qubit frequency variability and aging.” 7th paragraph
Overlap junctions fabricated in this work, however, show significantly less aging, limited to a decrease of ~~3.83.7%~~ over 146 days (Figure 4b).
- Supplementary Figure 2.

- Supplementary section “VIII. JJ normal resistance relative standard deviation analysis.”

One important remark to make here is that, according to model A, the contribution of the barrier non-uniformity to the measured resistance variability is dominant already for JJ with $A > 0.045$ $0.075 \mu\text{m}^2$, while according to model B, all measured structures are currently dominated by area variability.

- Table II.
- Supplementary Figure 8

Reviewer Reports on the First Revision:

Referees' comments:

Referee #1 (Remarks to the Author):

After reading the authors' responses to all of the reviewers' questions, I am not fully convinced that this manuscript is at the level of Nature publications. I acknowledge that the authors made significant revisions to their manuscript to address reviewers' comments and to improve the content, and that using semiconductor CMOS fabrication technology for superconducting qubits could be important for scaling up. I am still insisting that this work is a continuation of the authors' prior work published in npj Quantum Information in 2022. [1] I find the authors claim that "This is the very first demonstration of high coherence time qubits fabricated with a CMOS pilot line on a 300 mm wafer." to be inaccurate. The authors already made a similar claim in their previous work [1] that "The presented fabrication process, therefore, heralds an important milestone towards a manufacturable 300mm CMOS process for high-coherence superconducting qubits". Furthermore, the fabrication techniques, characterization of their qubits, and even the structure of the current manuscript resembles closely their earlier work. I was hoping to hear a convincing discussion from the authors about what is novel in their current manuscript.

This manuscript will possibly be (and should be) published. It is an editorial decision if Nature is the right platform.

[1] npj Quantum Information (2022) 8:93

Referee #2 (Remarks to the Author):

I thank the authors for responding to my comments. I think the responses were generally adequate. There is no technical debate here, as the authors have demonstrated an ability to perform 300-mm wafer-scale fab of superconducting qubits with moderate coherence (good, but not state-of-the-art). The blue-team/red-team evaluation I did before still stands. Whether this should be published in Nature is ultimately an editorial decision.

Referee #3 (Remarks to the Author):

Thank you for addressing the questions satisfactorily in the reply. I have a much better understanding of the manuscript and the data presentation. The authors have, in addition, discussed various ways in which they can improve, I appreciate it and I hope that the authors can share more of these in the main manuscript of the supplementary material.

Few minor issues, I still find some of the explanations in the main manuscript still lacking. Please improve the presentations on the following parts:

- Main manuscript, Section "Qubit coherence": "... Multiple loss channels contribute to the qubit relaxation times. A lower bound on the intrinsic JJ relaxation time of our process can be set by monitoring the.."

The authors assumed that the intrinsic loss contribution due to the JJ is the lowest among capacitive losses in the qubit. Unless it is obvious due to the way the authors designed the qubits, please write down such an assumption, so that it is clear that why the authors only consider the average T1 of the best performing qubits.

- The use of the term "intrinsic JJ relaxation time" can be confusing, I suggest to use another term that the authors used in the reply, such as: "JJ induced relaxation time", or "JJ-related relaxation time."

- Did the authors quote the value δ_t , i.e. the slope of fig. 3(a), somewhere in the manuscript. I fitted the data, and found a number in the order of $1e-3$, which doesn't sound too far from other numbers reported in the literature. Please specify the numbers for completeness

Congratulation to the authors for achieving this big milestone, I'm excited to see what is about to come for such an advanced foundry based fabrication of superconducting qubits!

Author Rebuttals to First Revision:

In general, we would like to thank all three referees again for their constructive feedback on our work. We are glad to read that our replies and revisions have been well received, and take also the second round of comments with much gratitude.

Our responses are written in blue.

Referee #1 (Remarks to the Author):

After reading the authors' responses to all of the reviewers' questions, I am not fully convinced that this manuscript is at the level of Nature publications. I acknowledge that the authors made significant revisions to their manuscript to address reviewers' comments and to improve the content, and that using semiconductor CMOS fabrication technology for superconducting qubits could be important for scaling up. I am still insisting that this work is a continuation of the authors' prior work published in *npj Quantum Information* in 2022. [1] I find the authors claim that "This is the very first demonstration of high coherence time qubits fabricated with a CMOS pilot line on a 300 mm wafer." to be inaccurate. The authors already made a similar claim in their previous work [1] that "The presented fabrication process, therefore, heralds an important milestone towards a manufacturable 300mm CMOS process for high-coherence superconducting qubits". Furthermore, the fabrication techniques, characterization of their qubits, and even the structure of the current manuscript resembles closely their earlier work. I was hoping to hear a convincing discussion from the authors about what is novel in their current manuscript.

This manuscript will possibly be (and should be) published. It is an editorial decision if Nature is the right platform.

[1] *npj Quantum Information* (2022) 8:93

We thank the referee for their honest opinion, and are happy to read that our revisions have convinced the referee of the importance of CMOS fabrication technology for superconducting qubit fabrication scaling.

Our previous work was an important derisking milestone that showed the process can "in principle" be setup in a fabrication facility. However, we respectfully disagree with the minimalization of this work as a mere "continuation". The earlier demonstration of a "fab compatible process" on coupons is not the same as actually fabricating wafers with yielding high-coherence qubits in an industry standard facility. These results are in fact the very first demonstration of high coherence qubits fabricated in a CMOS pilot line on a 300 mm wafer and are the culmination of many intensive new developments and optimizations on each step of the fabrication in CMOS tools. We are very excited about this milestone, a sentiment shared by many in the community, and hopefully the start of much more to come.

Referee #2 (Remarks to the Author):

I thank the authors for responding to my comments. I think the responses were generally adequate. There is no technical debate here, as the authors have demonstrated an ability to perform 300-mm wafer-scale fab of superconducting qubits with moderate coherence (good, but not state-of-the-art). The blue-team/red-team evaluation I did before still stands. Whether this should be published in Nature is ultimately an editorial decision.

We are happy to read that our replies to the referee's comments were adequate and we are grateful for their time and effort spent on improving this manuscript.

Referee #3 (Remarks to the Author):

Thank you for addressing the questions satisfactorily in the reply. I have a much better understanding of the manuscript and the data presentation. The authors have, in addition, discussed various ways in which they can improve, I appreciate it and I hope that the authors can share more of these in the main manuscript or the supplementary material.

We appreciate greatly the acknowledgement given by the referee. We are glad to read our first reply helped to clarify the results, we will take the additional comments to heart and improve the manuscript with the further suggestions.

Few minor issues, I still find some of the explanations in the main manuscript still lacking. Please improve the presentations on the following parts:

3.1 Main manuscript, Section "Qubit coherence": "... Multiple loss channels contribute to the qubit relaxation times. A lower bound on the intrinsic J relaxation time of our process can be set by monitoring the.."

The authors assumed that the intrinsic loss contribution due to the J is the lowest among capacitive losses in the qubit. Unless it is obvious due to the way the authors designed the qubits, please write down such an assumption, so that it is clear that why the authors only consider the average T1 of the best performing qubits.

We thank the referee for pointing out that the current phrasing is not entirely clear. The capacitor losses are dominant, but this is only revealed in the next section of the manuscript when we look at the scaling of the losses with the capacitor size. To avoid relying on results only discussed later in the manuscript, we have adjusted the motivation for plotting the best device relaxation times as follows:

"An upper bound on the qubit relaxation time obtained with our current process can be estimated by monitoring the time averaged relaxation times $\langle T_1 \rangle_t$ of the best performing qubits from sub-dies D across the wafer."

After which we discuss the observed centre-to-edge effect of this upper bound.

3.2 The use of the term "intrinsic J relaxation time" can be confusing, I suggest to use another term that the authors used in the reply, such as: " J induced relaxation time", or " J -related relaxation time."

We thank the referee for this suggestion, we agree and made the appropriate changes in the manuscript.

3.3 Did the authors quote the value δ_t , i.e. the slope of fig. 3(a), somewhere in the manuscript. I fitted the data, and found a number in the order of $1e-3$, which doesn't sound too far from other numbers reported in the literature. Please specify the numbers for completeness

We did not yet report the best fit value of δ_t in the manuscript. The referee is close in the quoted estimate. Our best fit value is $\delta_t = (1.8 \pm 0.2) \times 10^{-3}$, indeed within the ballpark of interface losses reported by others in literature. We did intentionally not include this value in the main manuscript because this value is directly dependent on to the interface participation ratio estimate. We used calculation methods and values for dielectric constants and thicknesses of the interface layers accepted by the community, however these are ultimately unknown. We are confident in the value of the product $\delta_t p_\Sigma$, however not in the value of p_Σ and therefore also not in the value of δ_t . In other words, we cannot exclude the situation $\delta_t p_\Sigma = a \delta_t p'_\Sigma = \delta'_t p'_\Sigma$. Additionally, our effective interface loss δ_t is some weighted sum of the three different interfaces (MA, MS and SA), so it is not directly representing a material loss tangent.

Despite our initial reservations about the relevance of this parameter, we have included it now in the methods section.

Congratulation to the authors for achieving this big milestone, I'm excited to see what is about to come for such an advanced foundry based fabrication of superconducting qubits!

Thank you very much!